# Interpreting the Weight Space
# of Customized Diffusion Models

**Amil Dravid**[*1,2]    **Yossi Gandelsman**[*1]    **Kuan-Chieh Wang**[2]

**Rameen Abdal**[3]    **Gordon Wetzstein**[3]    **Alexei A. Efros**[1]    **Kfir Aberman**[2]

[1]UC Berkeley  [2]Snap Inc.  [3]Stanford University

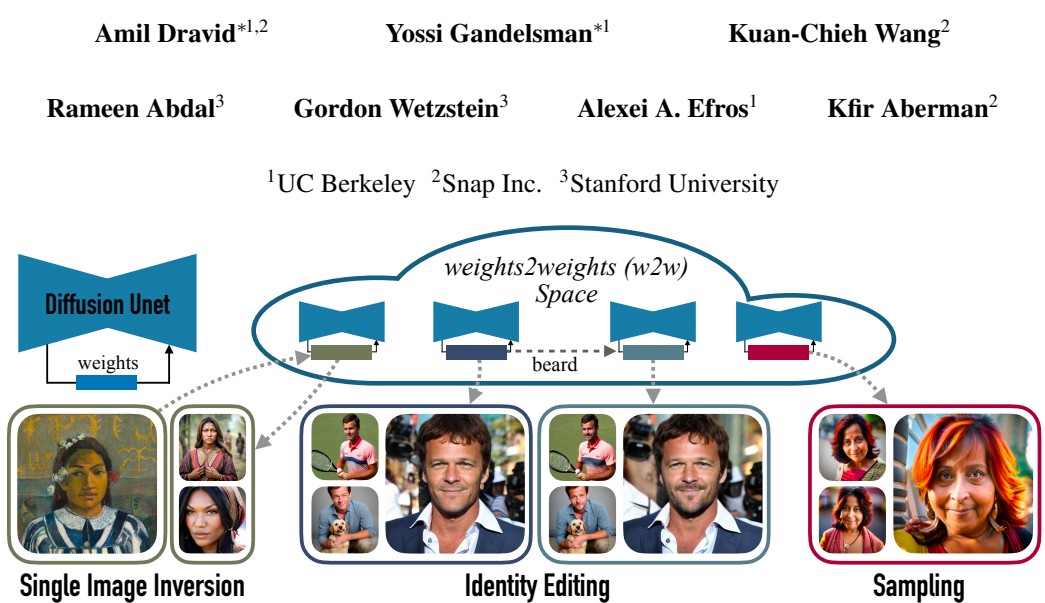

Figure 1: **weights2weights (w2w) space enables controllable creation of new customized diffusion models.** We model a manifold of customized diffusion models as a subspace of weights that encodes different instances of a broad visual concept (e.g., human identities, dog breeds, etc.). This forms a space that supports inverting the subject (e.g., identity) from a single image into a model, editing the subject encoded in the model, and sampling new models that encode new instances of the visual concept. Each of these operations results in a new model that can consistently generate the subject.

## Abstract

We investigate the space of weights spanned by a large collection of customized diffusion models. We populate this space by creating a dataset of over 60,000 models, each of which is a base model fine-tuned to insert a different person's visual identity. We model the underlying manifold of these weights as a subspace, which we term *weights2weights*. We demonstrate three immediate applications of this space that result in new diffusion models – sampling, editing, and inversion. First, sampling a set of weights from this space results in a new model encoding a novel identity. Next, we find linear directions in this space corresponding to semantic edits of the identity (e.g., adding a beard), resulting in a new model with the original identity edited. Finally, we show that inverting a single image into this space encodes a realistic identity into a model, even if the input image is out of distribution (e.g., a painting). We further find that these linear properties of the diffusion model weight space extend to other visual concepts. Our results indicate that the weight space of fine-tuned diffusion models can behave as an interpretable *meta*-latent space producing new models.[1]

---

[*]Equal contribution

[1]Project page: https://snap-research.github.io/weights2weights
 Code: https://github.com/snap-research/weights2weights

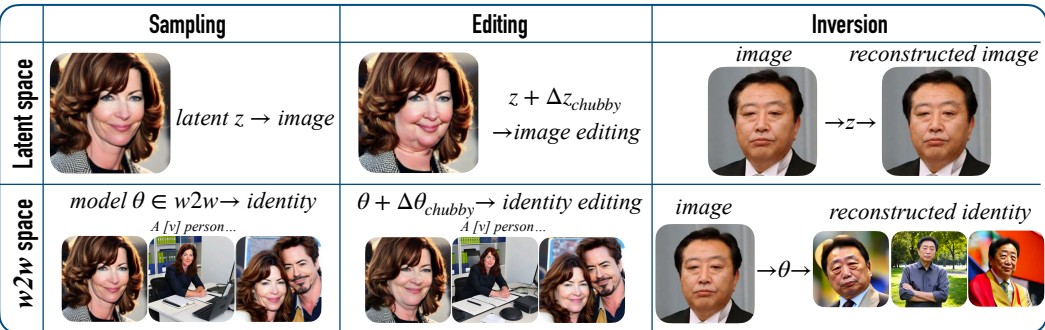

Figure 2: **The *weights2weights* space operates as a *meta*-latent space**. Unlike a traditional generative latent space, *w2w* space controls the model itself rather than single image instances. New identity-encoding models can be sampled from the space and edited by linearly traversing along semantic directions in weight space. Additionally, a single image can be inverted into the space to produce a model that consistently generates that identity.

# 1   Introduction

Generative models have emerged as a powerful tool to model our rich visual world. In particular, the latent space of single-step generative models, such as generative adversarial networks (GANs) [18, 28], has been shown to linearly encode meaningful concepts in the output images. For instance, datasets of latent vectors were used to discover linear directions in the GAN latent space encoding different attributes (e.g., gender or age of faces) [20, 60]. Even earlier, datasets of images and keypoints were leveraged to discover subspaces of facial shape and appearance [6, 53].

We aim to extend this even further, using datasets of model weights instead datasets of images or latents. Can we discover such interpretable subspaces in the model weights themselves? Recently introduced personalization approaches, such as Dreambooth [54] or Custom Diffusion [34], may hint that this is the case. These methods aim to learn an instance of a subject, such as a person's visual identity. Rather than searching for a latent code that represents an identity in the input noise space, these approaches customize diffusion models by fine-tuning on subject-specific images, which results in identity-specific model weights. We therefore hypothesize that a latent space can exist *in the weights themselves*.

To test our hypothesis, we fine-tune over 60,000 personalized models on individual identities to obtain points that lie on a manifold of customized diffusion model weights. To reduce the dimensionality of each data point, we use low-rank approximation (LoRA) [23] during fine-tuning and further apply Principal Components Analysis (PCA) to the set of data points. This forms our final space: *weights2weights* (*w2w*). Unlike traditional generative models like GANs, which model the pixel space of images, we model the *weight space* of these personalized models. Thus, each sample in our space corresponds to an identity-specific model which can consistently generate that subject. We provide a schematic in Fig. 2 that contrasts a typical latent space with our proposed *w2w* space, demonstrating the differences and analogies between these two representations. *w2w* space can be thought of as a *meta*-latent space, enabling controllable creation of new models instead of just images like a traditional latent space.

Creating this space unlocks a variety of applications that involve traversal in *w2w* (Fig. 1). First, we demonstrate that sampling model weights from *w2w* space corresponds to a new model encoding a novel subject. Second, we find linear directions in this space corresponding to semantic edits of the identity. Finally, we show that enforcing weights to live in this space enables a diffusion model to learn a subject given a single image, even if it is out of distribution.

We find that *w2w* space is highly expressive through quantitative evaluation on editing customized models and encoding new identities given a single image. Qualitatively, we observe this space supports sampling models that encode diverse and realistic identities, while also capturing the key characteristics of out-of-distribution identities. We finally demonstrate that similar weight subspaces exist for other visual concepts such as dog breeds and car types.

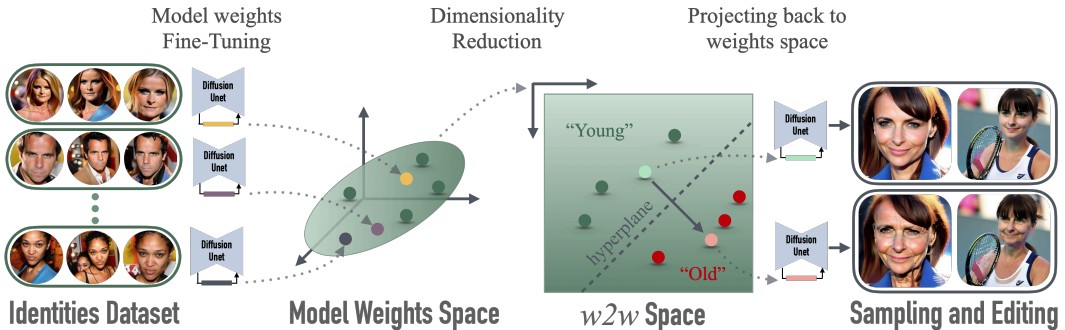

| Model weights Fine-Tuning | Dimensionality Reduction | Projecting back to weights space | |
|---|---|---|---|
| **Identities Dataset** | **Model Weights Space** | *w2w* **Space** | **Sampling and Editing** |

Figure 3: **Building *weights2weights* (*w2w*) space.** We create a dataset of model weights where each model is personalized to a specific identity using low-rank updates (LoRA). These model weights lie on a weights manifold that we further project into a lower-dimensional subspace spanned by its principal components. We train linear classifiers to find disentangled edit directions in this space.

## 2   Related Work

**Image-based generative models.** Various models have been proposed for image generation, including variational autoencoders (VAEs) [31], flow-based models [12, 32, 49], generative adversarial networks (GANs) [18], and diffusion models [22, 43, 62]. Within the realm of high-quality photo-realistic image generation, GANs [25, 28, 29] and diffusion models [22, 43, 52, 63] have garnered significant attention due to their controllability and ability to produce high-quality images. Leveraging the compositionality of these models, methods for personalization and customization have been developed which aim to insert user-defined concepts via fine-tuning [16, 34, 40, 54]. Various works try to reduce the dimensionality of the optimized parameters for personalization either by operating in specific model layers [34] or in text-embedding space [16], by training hypernetworks [55], and by constructing a linear basis in text embedding space [73].

**Latent space of generative models.** Linear latent space models of facial shape and appearance were studied extensively in the 1990s, using PCA-based representations (e.g. Active Appearance Models [9], 3D Morphable Models [6]) as well as operating directly in pixel and keypoint space [53]. However, these techniques were restricted to aligned and cropped frontal faces. More recently, generative adversarial networks (GANs), particularly the StyleGAN series [26, 27, 28, 29], have showcased editing capabilities facilitated by their interpretable latent space. Furthermore, linear directions can be found in their latent space to conduct semantic edits by training linear classifiers or applying PCA [20, 60], among other methods for discovering semantic directions [8, 66]. Several methods aim to project real images into the GAN latent space in order to conduct this editing [1, 3, 51, 64, 77]. Beyond the latent space, works such as [4] found that directions could be discovered in the neuron activation space, suggesting the interpretability of weights.

Although diffusion models architecturally lack a GAN-like latent space, some works aim to discover similar spaces in these models. This has been explored in the UNet bottleneck layer [35, 41], noise space [10, 78], and text-embedding space [5]. Concept Sliders [17] explores the weight space for semantic image editing by conducting low-rank training with contrasting image or text pairs.

**Weights as data.** Past works have exploited the structure within weight space of deep networks for various applications. In particular, some have found linear properties of weights, enabling simple model ensembling and editing via arithmetic operations [24, 56, 59, 69]. Other works create datasets of neural network parameters for training hypernetworks [14, 19, 42, 55, 68], predicting properties of networks [58], and creating design spaces for models [46, 47].

## 3   Method

We start by demonstrating how we create a manifold of model weights as illustrated in Fig. 3. We explain how we obtain low-dimensional data points for this space, each of which represents an individual subject from a broad class (i.e., identity). We then use these points to model a weights manifold. Next, we find linear directions in this manifold that correspond to semantic attributes and use them for editing the identities. Finally, we demonstrate how this manifold can be utilized for constraining an ill-posed inversion task with a single image to reconstruct its identity.

## 3.1 Preliminaries

In this section, we first introduce latent diffusion models (LDM) [52], which we will use to create a dataset of weights. Then, we explain the approach for obtaining identity-specific models from LDM via Dreambooth [54] fine-tuning. We finally present a version of fine-tuning that uses low-dimensional weight updates (LoRA [23]). We will use the fine-tuned low-dimensional per-identity weights as data points to construct the weights manifold in Sec. 3.2.

**Latent diffusion models [52].** We will extract weights from latent diffusion models to create *w2w* space. These models follow the standard diffusion objective [22] while operating on latents extracted from a pre-trained Variational Autoencoder [15, 31, 50]. With text, the conditioning signal is encoded by a text encoder (such as CLIP [44]), and the resulting embeddings are provided to the denoising UNet model. The loss of latent diffusion models is:

$$\mathbb{E}_{\mathbf{x},\mathbf{c},\epsilon,t}[w_t||\epsilon - \epsilon_\theta(\mathbf{x}_t,\mathbf{c},t)||_2^2], \tag{1}$$

where $\epsilon_\theta$ is the denoising UNet, $\mathbf{x}_t$ is the noised version of the latent for an image, $\mathbf{c}$ is the conditioning signal, $t$ is the diffusion timestep, and $w_t$ is a time-dependent weight on the loss.

To sample from the model, a random Gaussian latent $x_T$ is deterministically denoised conditioned on a prompt for a fixed set of timesteps with a DDIM sampler [63]. The denoised latent is then fed through the VAE decoder to generate the final image.

**Dreambooth [54].** To obtain an identity-specific model, we use the Dreambooth personalization method. Dreambooth fine-tuning introduces a novel subject into a pre-trained diffusion model given only a few images of it. During training, Dreambooth follows a two-part objective:

$$\mathbb{E}_{\mathbf{x},\mathbf{c},\epsilon,t}[w_t||\epsilon - \epsilon_\theta(\mathbf{x}_t,\mathbf{c},t)||_2^2 + \lambda w_{t'}||\epsilon' - \epsilon_\theta(\mathbf{x}_t',\mathbf{c}',t')||_2^2], \tag{2}$$

where the first term corresponds to the standard diffusion denoising objective using the subject-specific data $\mathbf{x}$ conditioned on the text prompt "[identifier] [class noun]" (e.g., "*[v] person*"), denoted $\mathbf{c}$. The second term, weighted by $\lambda$, corresponds to a prior preservation loss, which involves the standard denoising objective using the model's own generated samples $\mathbf{x}'$ for the broader class $\mathbf{c}'$ (e.g., "person"). This prevents the model from associating the class name with the specific instance, while also leveraging the semantic prior on the class.

**Low Rank Adaptation (LoRA) [23].** Dreambooth requires fine-tuning all the weights of a model, which is a high–dimensional space. We turn to a more efficient fine-tuning scheme, LoRA, that modifies only a low-rank version of the weights. LoRA uses weight updates $\Delta W$ with a low intrinsic rank. For a base model layer $W \in \mathbb{R}^{m \times n}$, the LoRA update for that layer $\Delta W$ can be decomposed into $\Delta W = BA$, where $B \in \mathbb{R}^{m \times r}$ and $A \in \mathbb{R}^{r \times n}$ are low-rank matrices with $r \ll min(m,n)$. During training, for each model layer, only the $A$ and $B$ are updated. This significantly reduces the number of trainable parameters. During inference, the low-rank weights are added residually to the weights of each layer in the base model and scaled by a coefficient $\alpha \in \mathbb{R}$: $W + \alpha\Delta W$.

## 3.2 Constructing the weights manifold

**Creating a dataset of model weights.** To construct the *weights2weights* (*w2w*) space, we begin by creating a dataset of model weights $\theta_i$. We conduct Dreambooth fine-tuning on latent diffusion models in order to insert new subjects with the ability to control image instances using text prompts. This training is done with LoRA in order to reduce the space of model parameters. Each model is fine-tuned on a set of images corresponding to one human subject. After training, we flatten and concatenate all of the LoRA matrices, resulting in a data point $\theta_i \in \mathbb{R}^d$ which represents one identity. After training over $N$ different instances, we have our final dataset of model weights $\mathcal{D} = \{\theta_1, \theta_2, ..., \theta_N\}$, representing a diverse array of subjects.

**Modeling the weights manifold.** We posit that our data $D \subseteq \mathbb{R}^d$ lies on a lower-dimensional manifold of weights that encode identities. A randomly sampled set of weights in $\mathbb{R}^d$, would not be guaranteed to produce a valid model encoding identity as the $d$ degrees of freedom can be fine-tuned for any purpose. Therefore, we hypothesize that this manifold is a subset of the weight space. Inspired by findings that high-level concepts can be encoded as linear subspaces of representations [13, 37, 45, 48], we model this subset as a linear subspace $\mathbb{R}^m$ where $m < d$, and call it *weights2weights* (*w2w*) space. We represent points in this subspace as a linear combination

of basis vectors $\mathbf{w} = \{w_1, ..., w_m\}$, $w_i \in \mathbb{R}^d$. In practice, we apply Principal Component Analysis (PCA) on the $N$ models and keep the first $m$ principal components for dimensional reduction and forming our basis of $m$ vectors.

**Sampling from the weights manifold.** After modeling this weights manifold, we can sample a new model that lies on it, resulting in a new model that generates a novel identity. We sample a model represented with basis coefficients $\{\beta_1, ..., \beta_m\}$, where each coefficient $\beta_k$ is sampled from a normal distribution with mean $\mu_k$ and standard deviation $\sigma_k$. The mean and standard deviation are calculated for each principal component $k$ from the coefficients among all the training models.

### 3.3 Finding Interpretable Weight Space Directions

We seek a direction $\mathbf{n} \in \mathbb{R}^d$ defining a hyperplane that separates between binary identity properties embedded in the model weights (e.g., male/female), similarly to hyperplanes observed in the latent space of GANs [60]. We assume binary labels are given for attributes present in the identities encoded by the models. We then train linear classifiers using weights of the models as data based on these labels, imposing separating hyperplanes in weight space. Given an identity parameterized by weights $\theta$, we can manipulate a single attribute by traversing in a direction $\mathbf{n}$, orthogonal to the separating hyperplane: $\theta_{\text{edit}} = \theta + \alpha\mathbf{n}$. An edit operation in *w2w* space produces a new model with the original subject edited, allowing the model to generate infinitely many new images of the edited subject.

### 3.4 Inversion into *w2w* Space

Traditionally, inversion of a generative model involves finding an input such as a latent code that best reconstructs a given image [38, 70]. This corresponds to finding a projection of the input onto the learned data manifold [77]. With *w2w* space, we model a manifold of model weights rather than images. Inspired by latent optimization methods [1, 77], we propose a gradient-based method of inverting a single identity from an image into our discovered space.

Given a single image $\mathbf{x}$, we follow a constrained denoising objective:

$$\max_{\theta} \mathbb{E}_{\mathbf{x},\mathbf{c},\epsilon,t}[w_t||\epsilon - \epsilon_\theta(\mathbf{x}_t, \mathbf{c}, t)||_2^2] \quad \text{s.t. } \theta \in \textit{w2w} \tag{3}$$

Specifically, we constrain the model weights to lie in *w2w* space by optimizing a set of basis coefficients $\{\beta_1, ..., \beta_m\}$ rather than the original parameters. Unlike Dreambooth, we do not employ a prior preservation loss, since the optimized model lies in the subspace defined by our dataset of weights, and inherits their priors.

## 4 Experiments

We demonstrate *w2w* space on the visual concept of human identities for a variety of applications. We begin by detailing implementation details. Next, we use *w2w* space for 1) sampling new models encoding novel identities, 2) editing identity attributes in a consistent manner via linear traversal in *w2w* space, 3) embedding a new identity given a single image, and 4) projecting out-of-distribution identities into *w2w* space. Finally, we analyze how scaling the number of models in our dataset of model weights affects the disentanglement of attribute directions and preservation of identity.

### 4.1 Implementation Details

**Creating an identity dataset.** We generate a synthetic dataset of ∼65,000 identities using [67], where each identity is associated with multiple images of that person. Each identity is based on an image with labeled binary attributes (e.g., male/female) from CelebA [36]. Each set of images corresponding to an identity is then used as data to fine-tune a latent diffusion model with Dreambooth. Further details on this dataset and train/test splits are provided in Appendix E.

**Encoding identities into model weights.** We conduct Dreambooth fine-tuning using LoRA with rank 1 on the identities. Following [56], we only fine-tune the query and value projection matrices in the cross-attention layers. We utilize the RealisticVision-v51 checkpoint[2] based on Stable Diffusion

---

[2] https://huggingface.co/stablediffusionapi/realistic-vision-v51

1.5. Conducting Dreambooth fine-tuning on each identity training set results in a dataset of ∼65,000 weights $\theta$ where $\theta \in \mathbb{R}^{100,000}$.

**Finding semantic attribute directions.** We utilize binary attribute labels from CelebA to train linear classifiers on the dataset of model weights we curated. We run Principal Component Analysis (PCA) on the ∼65,000 training models and project to the first 1000 principal components in order to reduce the dimensionality. The orthogonal edit directions are calculated via the analytic least squares solution on the matrix of projected training models $\mathcal{D} \in \mathbb{R}^{65,000 \times 1000}$, and then unprojected to the original dimensionality of the model weights: $\theta \in \mathbb{R}^{100,000}$.

## 4.2 Sampling from *w2w* Space

We present images generated from models that were sampled from the weights manifold (i.e., *w2w* Space) in Fig. 4. We follow the sampling procedure from Sec. 3.2, and generate images from the sampled model. As shown, each new model encodes a novel, realistic, and consistent identity. Additionally, we present the nearest neighbor model among the training dataset of model weights. We use cosine similarity on the models' principal component representations. Comparing with the nearest neighbors shows that these samples are not just copies from the dataset, but rather encode diverse identities with different attributes. Yet, the samples still demonstrate some similar features to the nearest neighbors. These include jawline and eye shape (top row), facial hair (middle row), and nose and eye shape (bottom row). Appendix A includes more such examples.

| Sampled Identity | Nearest Neighbor |
| --- | --- |

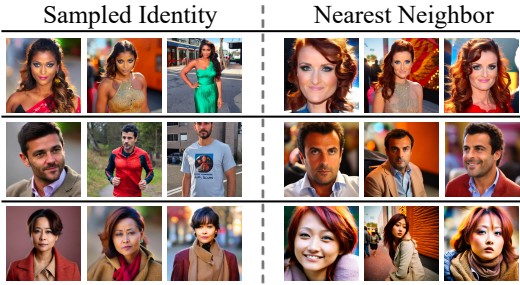

Figure 4: **Identity samples from *w2w* space.** We show the samples from *w2w* space do not overfit to nearest-neighbor identities, although they incorporate facial attributes from them. The identities are diverse and consistent across generations.

## 4.3 Editing Subjects

We demonstrate how directions found by the linear classifiers can be used to edit subjects encoded in the models. It is desired that these edits are 1) disentangled (i.e., do not interfere with other attributes of the embedded subject and preserve all other concepts such as context) 2) identity preserving (i.e., the person is still recognizable) 3) and semantically aligned with the intended edit.

**Baselines.** We compare against a naïve baseline of prompting with the desired attribute (e.g., "*[v]* person with small eyes"), and then Concept Sliders [17], an instance-specific editing method which we adapt to subject editing. In particular, we train their most accessible method, the text-based slider, which trains LoRAs to modulate attributes in a pretrained diffusion model based on contrasting text prompts. We then apply these sliders to the personalized identity models.

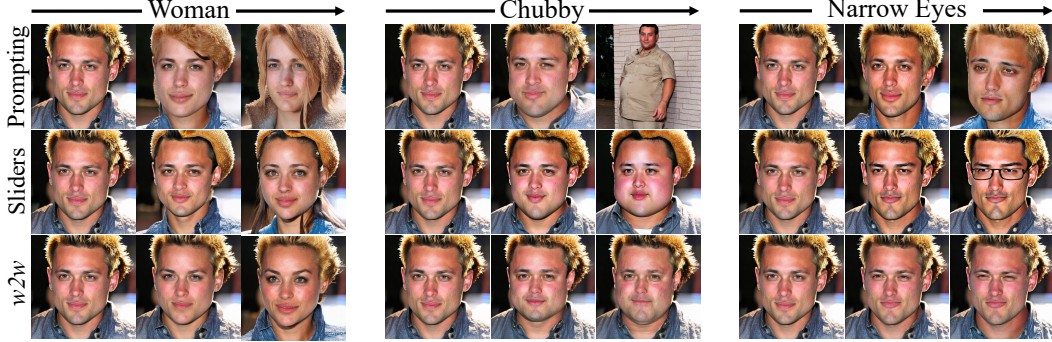

Figure 5: **Qualitative comparison.** *w2w* edits preserve identity while being disentangled and semantically aligned. Concept Sliders [17] tends to exaggerate effects which induces artifacts and degrades identity, while prompting the subject with the desired edit has unexpected effects.

Table 1: **Edits in *w2w* space preserve identity, are disentangled, and semantically aligned.**

| | ID Score ↑ | | | LPIPS ↓ | | | CLIP Score ↑ | | |
| | Prompting | Sliders | *w2w* | Prompting | Sliders | *w2w* | Prompting | Sliders | *w2w* |
|---|---|---|---|---|---|---|---|---|---|
| Gender | $0.39_{\pm 0.08}$ | $0.33_{\pm 0.09}$ | $\textbf{0.45}_{\pm 0.09}$ | $\textbf{0.30}_{\pm 0.05}$ | $0.39_{\pm 0.09}$ | $0.31_{\pm 0.03}$ | $1.98_{\pm 0.78}$ | $3.50_{\pm 0.68}$ | $\textbf{4.13}_{\pm 0.59}$ |
| Chubby | $0.29_{\pm 0.14}$ | $0.33_{\pm 0.09}$ | $\textbf{0.45}_{\pm 0.09}$ | $0.41_{\pm 0.05}$ | $0.38_{\pm 0.04}$ | $\textbf{0.36}_{\pm 0.04}$ | $1.12_{\pm 0.61}$ | $\textbf{2.21}_{\pm 0.61}$ | $2.16_{\pm 0.51}$ |
| Eyes | $0.52_{\pm 0.06}$ | $0.53_{\pm 0.04}$ | $\textbf{0.72}_{\pm 0.05}$ | $0.32_{\pm 0.03}$ | $0.30_{\pm 0.02}$ | $\textbf{0.19}_{\pm 0.02}$ | $0.17_{\pm 0.17}$ | $0.01_{\pm 0.22}$ | $\textbf{0.59}_{\pm 0.19}$ |

Figure 6: **Composing edits in *w2w* space.** Each column represents fixed seed samples from an edited model. Multiple edits in *w2w* space minimally degrade the original identity or interfere with other concepts, while maintaining edit appearance across different samples.

**Evaluation protocol.** We evaluate these three methods for identity preservation, disentanglement, and edit coherence. To measure identity preservation, we first detect faces in the original generated images and the result of the edits using MTCNN [74]. We then calculate the similarity of the FaceNet [57] embeddings. We also use LPIPS [76] computed between the images before and after the edit to measure the degree of disentanglement with other visual elements, and CLIP score [21], to measure if the desired edit matches the text caption for the edit.

To generate samples, we fix a set of prompts and random seeds which are used as input to the held-out identity models. Then, we choose a set of identity-specific manipulations. For prompt-based editing, we augment the attribute description to the set of fixed prompts (e.g., "chubby *[v]* person"). For Concept Sliders and *w2w*, we apply the weight space edit directions to the personalized model with a fixed norm which determines the edit strength. The norm is calculated using the maximum projection component onto the edit direction among the training set of model weights.

***w2w* edits are identity preserving and disentangled.** We evaluate over a range of identity-specific attributes and present three (gender, chubby, narrow eyes) in Tab. 1. Edits in *w2w* preserve the identity of the original subject as measured by the ID score. These edits are semantically aligned with the desired effect as indicated by the CLIP score while minimally interfering with other visual concepts, as measured by LPIPS. We note that the CLIP score can be noisy in this setting as text captions can be too coarse to describe attributes as nuanced as those related to the human face. We supplement this with a user study presented in Appendix B.

Qualitatively, *w2w* edits make the minimal amount of changes to achieve semantic and identity-preserving edits (Fig. 5). For instance, changing the gender of the man does not significantly change the facial structure or hair, unlike Concept Sliders or prompting with text descriptions. Prompting has inconsistent results, either creating no effect or making drastic changes. Concept Sliders tends to make caricaturized effects, such as making the man cartoonishly chubby and baby-like.

**Composing edits.** Edit directions in *w2w* space can be composed linearly as shown in Fig. 6. The first column represents samples from the original model, and each subsequent column represents samples from the edited models. Each row shares the same fixed random generation seed. The composed edits persist in appearance across different generations, binding to the identity. Furthermore, the edited weights result in a new model, where the subject has different attributes while still maintaining as much of the prior identity. This is in contrast to editing in a traditional latent space, where an edit only corresponds to a single image. Additionally, as we operate on an personalized identity-specific weight manifold, minimal changes are made to other concepts, such as scene layout or other people. For instance, in Fig. 6, adding edits to the woman does not interfere with the person standing by her.

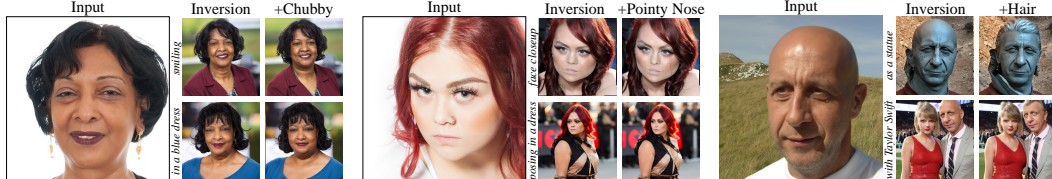

Figure 7: **Single image inversion reconstructs identity and enables editing in *w2w* space.** We present generated samples from the inverted models. These inverted identities can be composed in novel contexts and edited using our discovered semantic directions in weight space. These edits persist in appearance across generation seeds and prompts.

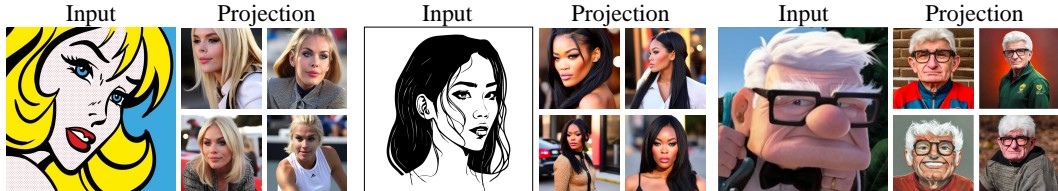

Figure 8: **Projecting out-of-distribution identities.** We show that our inversion method can convert unrealistic identities into realistic renderings with in-domain facial features. Each image represents a generated sample from the inverted model. The resulting identities can be composed in novel scenes, such as playing tennis or rendered into other artistic domains.

## 4.4 Inverting Subjects

**Evaluation protocol.** We measure *w2w* space's ability to represent novel identities by inverting a set of 100 random FFHQ [28] face images. We follow our inversion objective from eq. 3. We then provide a set of diverse prompts to generate multiple images and follow the identity preservation metric from Sec. 4.3 to measure subject fidelity. Implementation details are provided in Appendix C.

We compare our results to two approaches that use Dreambooth with rank-1 LoRA. The first is trained on a single image. The second is trained on *multiple images of each identity*. We generate such images by following our identity dataset construction from Sec. 4.1. This approach can be viewed as a pseudo-upper bound on modeling identity as it uses multiple images.

***w2w* space provides a strong identity prior.** Inverting a single image into *w2w* space improves on the single image Dreambooth baseline and closes the gap with the Dreambooth baseline that uses multiple identity images (Tab. 2). Conducting Dreambooth fine-tuning with a single image in the original weight space leads to image overfitting and poor subject reconstruction as indicated by a lower ID score. In contrast, by constraining the optimized weights to lie on a manifold of identity weights, *w2w* inversion inherits the rich priors of the models used to discover the space. As such, it can extract a high-fidelity identity that is consistent and compositional across generations. We present qualitative comparisons against Dreambooth and single-image Dreambooth in Appendix C. We additionally compare against other personalization methods in that section.

**Inverted models are editable.** Fig. 7 demonstrates that a diverse set of identities can be faithfully represented in *w2w* space. After inversion, the encoded identity can be composed in novel contexts and poses. For instance, the inverted man (rightmost example) can be seen posing with a celebrity or rendered as a statue. Moreover, semantic edits can be applied to the inverted models while maintaining appearance across generations.

Table 2: ***w2w* Inversion closes the gap with Dreambooth.**

| Method | Single-Image | ID Score ↑ |
|---|---|---|
| DB-LoRA | × | **0.69** ± 0.01 |
| DB-LoRA | ✓ | 0.43 ± 0.03 |
| *w2w* | ✓ | 0.64 ± 0.01 |

## 4.5 Out-of-Distribution Projection

***w2w* space captures out-of-distribution identities.** We follow the *w2w* inversion method from Sec. 4.4 to project images of unrealistic identities (e.g., paintings, cartoons, etc.) onto the weights manifold, and present these qualitative results in Fig. 8. By constraining the optimized model to live in *w2w* space, the inverted identities are converted into realistic renditions of the stylized identities, capturing prominent facial features. In Fig. 8, notice how the inverted identities generate a similar

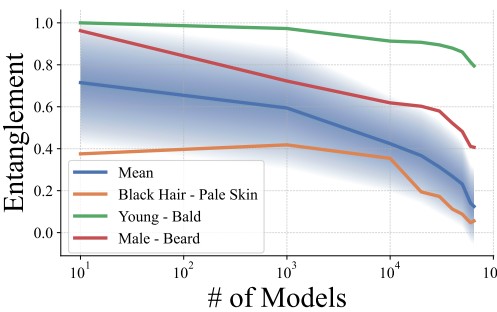 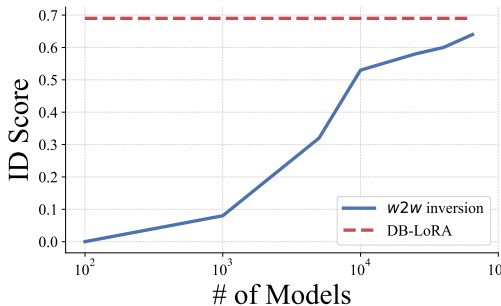

Figure 9: **Scaling dataset of models further disentangles classifier directions.** We highlight the trend in disentanglement of three examples where attributes may be strongly correlated among identities. As the number of models is increased, the features are less entangled.

Figure 10: **Scaling the number of models improves identity preservation.** As the span of *w2w* space increases, inversion can reconstruct single-image identities more faithfully, approaching the pseudo-upper bound of multi-image Dreambooth (DB-LoRA).

blonde hairstyle and nose structure in the first example, defined jawline and lip shape in the second example, and head shape and big nose in the last example. As also shown in the figure, the inverted identities can also be translated to other artistic domains using text prompts. We present a variety of domains projected into *w2w* space in Appendix D.

## 4.6 Effect of Number of Models Spanning *w2w* Space

We ablate the number of models used to create *w2w* space and investigate the expressiveness of the resulting space. In particular, we measure the degree of entanglement among the edit direction and how well this space can capture identity.

**Disentanglement vs. the number of models.** We find that scaling the number of models in our dataset of weights leads to less entangled edit directions in *w2w* space (Fig. 9). We vary the number of models in our dataset of weights and reapply PCA to establish a basis. We then measure the absolute value of cosine similarity (lower is better) between all pairs of linear classifier directions found for CelebA labels. We repeat this as we scale the number of model weights used to train the classifiers. We report the mean and standard deviation for these scores, along with three notable semantic direction pairs. We observe a trend in decreasing cosine similarity. Notably, pairs such as "Black Hair - Pale Skin," "Young - Bald," and "Male - Beard" which may correlate in the distribution of identities, become less correlated as we scale our dataset of model weights.

**Identity preservation vs. the number of models.** We observe that as we scale the number of models in our dataset of weights, identities are more faithfully represented in *w2w* space (Fig. 10). We follow the same procedure as the disentanglement ablation, reapplying PCA to establish a basis based on the dataset of model weights. Next, following Sec. 4.4, we optimize coefficients for this basis and measure the average ID score over the 100 inverted FFHQ evaluation identities. As each model in our dataset encodes a different instance of an identity, growing this dataset increases the span of *w2w* space and its ability to capture more diverse identities. We plot the average multi-image Dreambooth LoRA (DB-LoRA) ID score from Sec. 4.4, which is agnostic to our dataset of models. This establishes a pseudo-upper bound on identity preservation. Scaling enables *w2w* to represent identities given a single image with performance approaching that of traditional Dreambooth with LoRA, which uses multiple images and trains in a higher dimensional space.

## 5 Extending to Other Domains

We extend our hypothesis of interpretable linear weight subspaces in diffusion models to other visual concepts beyond human identities. We apply the *weights2weights* framework to form a subspace for models encoding different dog breeds. To create a dataset for fine-tuning, we generate images with Stable Diffusion based on each of the 120 dog classes from ImageNet [11]. We then conduct Dreambooth fine-tuning on each set of dog breed images to create a dataset of 120 dog-encoding models, subsequently applying PCA. To find edit directions, we use GPT-4 [2] to create labels for

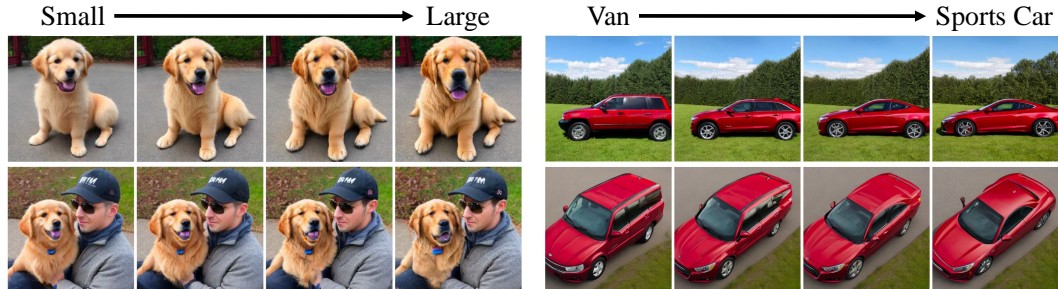

Figure 11: *weights2weights* **linear subspaces can be created for other visual concepts.** We follow the same procedure of applying PCA and finding edit directions with linear classifiers on datasets of models encoding dog breeds and models encoding car types.

each dog breed (e.g., wavy hair or not) and then train linear classifiers on the model weight principal component projections like Sec. 3.3. We apply the same framework to different car categories using models fine-tuned on images from a dataset of 197 different car types [33]. We present results for traversing edit directions in these two subspaces in Fig. 11. Each column represents samples from an edited model. Each row shares the same fixed random generation seed.

Our results provide further evidence that diffusion models can encode visual concepts linearly. This enables the creation of new models in a controlled manner via simple interpolation. For instance, in Fig. 11, we rewrite the model's learned concept of a small golden retriever to make it bigger, or the model's encoding of a red van to make it a sports car. Additionally, unlike older PCA-based methods [6, 53, 61, 65] which rely on aligned pixels or keypoints of human faces, *weights2weights* can extend to other domains beyond human identities. We refer the reader to Appendix G for more results of applying *w2w* space to other visual concepts.

## 6   Limitations

As with any data-driven method, *w2w* space inherits the biases of the data used to discover it. For instance, co-occurring attributes in the identity-encoding models would cause linear classifier directions to entangle them (e.g. gender and facial hair). However, as we scale the number of models, spurious correlations will drop as evidenced by Fig. 9. These semantic directions are also limited by the labels present in CelebA. Additionally, the span of the *w2w* space is dictated by the models used to create it. Thus, *w2w* space can struggle to represent more complex identities as seen in Fig. 12. Inversion

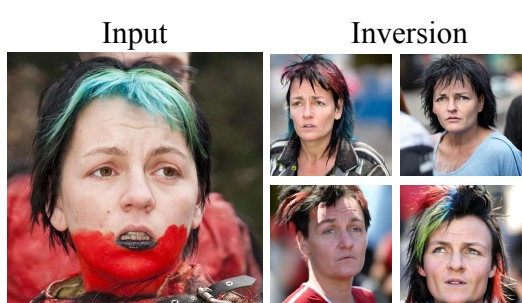

Figure 12: *weights2weights* **fails to capture identities with undersampled attributes.**

in these cases amounts to projecting onto the closest identity on the weights manifold. Despite this, our analysis on the size of the model dataset reveals that forming a space using a larger and more diverse set of identity-encoding models can mitigate this limitation.

## 7   Discussion and Broader Impact

We presented a paradigm for representing diffusion model weights as a point in a subspace defined by other customized models – *weights2weights* (*w2w*) space. This enabled applications analogous to those of a generative latent space – inversion, editing, and sampling – but producing model weights rather than images, resulting in what we term a *meta*-latent space. We demonstrated these applications on model weights encoding human identities and extended this space to other visual concepts. Although these applications could enable malicious manipulation of real human identities and model weights, we hope the community uses the framework to explore visual creativity as well as utilize this interpretable space for controlling models for safety.

## Acknowledgements

The authors would like to thank Grace Luo, Lisa Dunlap, Konpat Preechakul, Sheng-Yu Wang, Stephanie Fu, Or Patashnik, Daniel Cohen-Or, and Sergey Tulyakov for helpful discussions. AD is supported by the US Department of Energy Computational Science Graduate Fellowship. Part of the work was completed by AD as an intern with Snap Inc. YG is funded by the Google Fellowship. Additional funding came from ONR MURI.

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

# A  Sampling

We present additional examples of models sampled from *w2w* space in Fig. 13. The sampled models encode a diverse array of identities which are not copied from the dataset of model weights, as seen by comparing them to the nearest neighbor models from the training set. However, there are attributes borrowed from the nearest neighbors which visually appear in the sampled identity. For instance, the sampled man in the first row shares a similar jawline to the nearest neighbor identity. The sampled identities also demonstrate the same ability as the original training identities to be composed into novel contexts. A variety of prompts are used in Fig. 13 and the identities are still consistent.

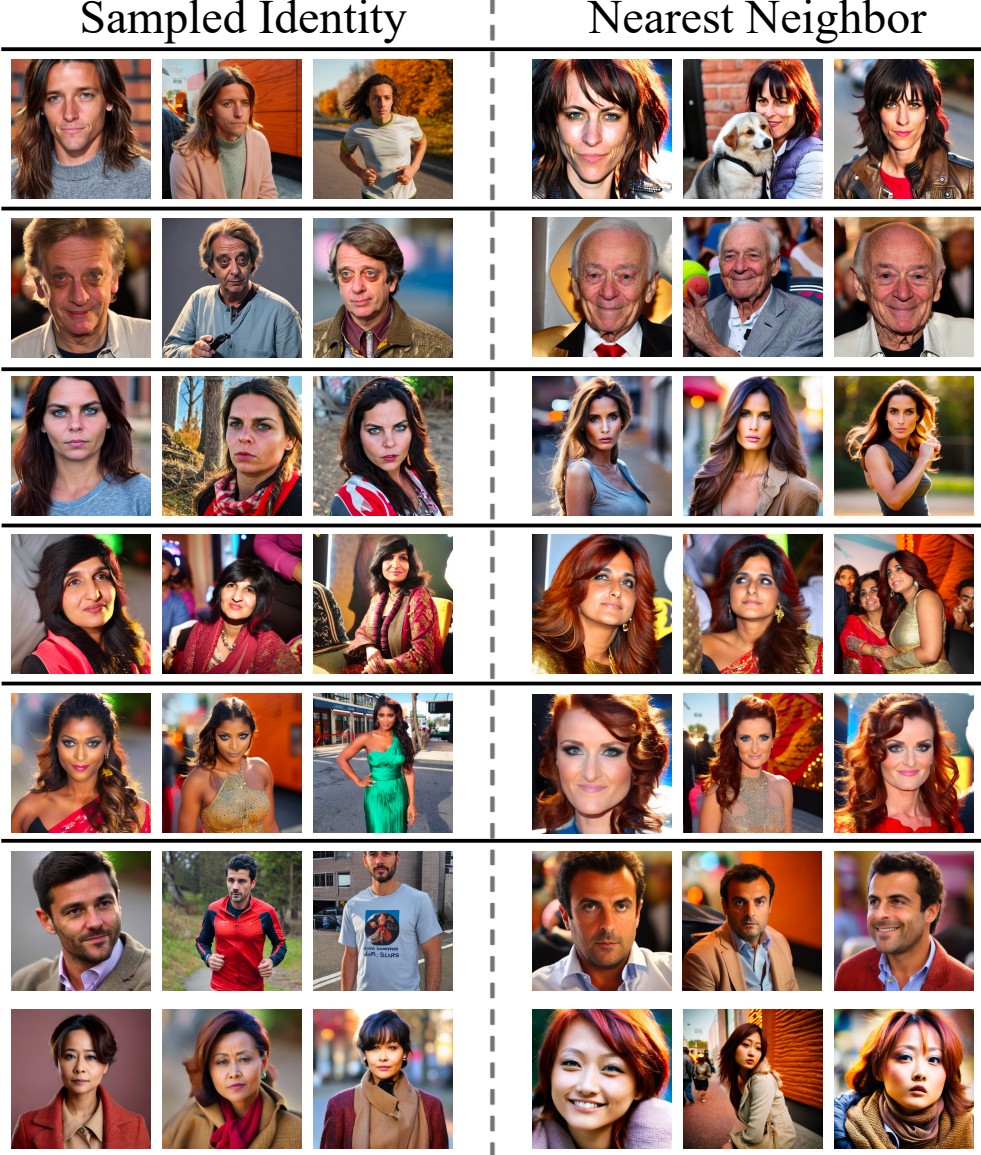

Figure 13: **Sampled identity-encoding models from *w2w* space and their nearest neighbor models.** The sampled identities share some characteristics with the nearest neighbors, but are still distinct. These identities can also be composed into novel contexts like a standard customized diffusion model.

# B    Model Editing in *w2w* Space

**Qualitative Results.** We display additional examples of applying edits in *w2w* space based on the directions discovered using linear classifiers and CelebA labels. In Fig. 14, we demonstrate how the strength of these edits can be modulated and combined with minimal interference. These edits are apparent even in more complex scenes beyond face images. Also, the edits do not degrade other present concepts, such as the dog near the man (top left example).

In Figs. 15 and 16, we demonstrate how multiple edits can be progressively added in a disentangled fashion with minimal degradation to the identity. Additionally, since we operate in a subspace of weight space, these edits persist with a consistent appearance across different generations. For instance, even the man exhibits the edits as a painting in Fig. 15.

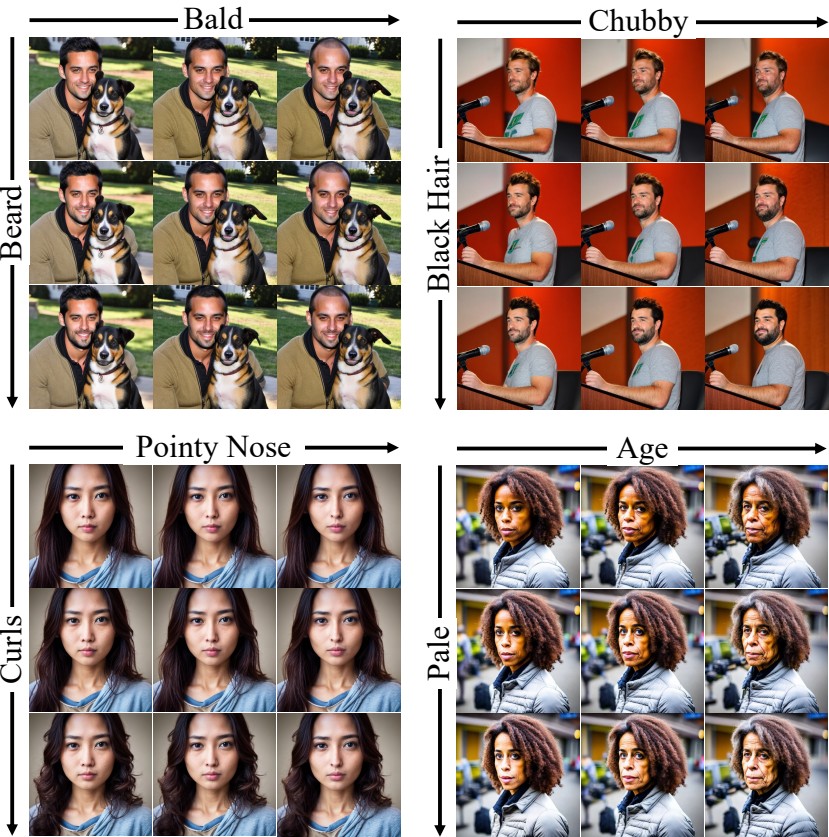

Figure 14: **Multiple edits can be controlled in a continuous manner.**

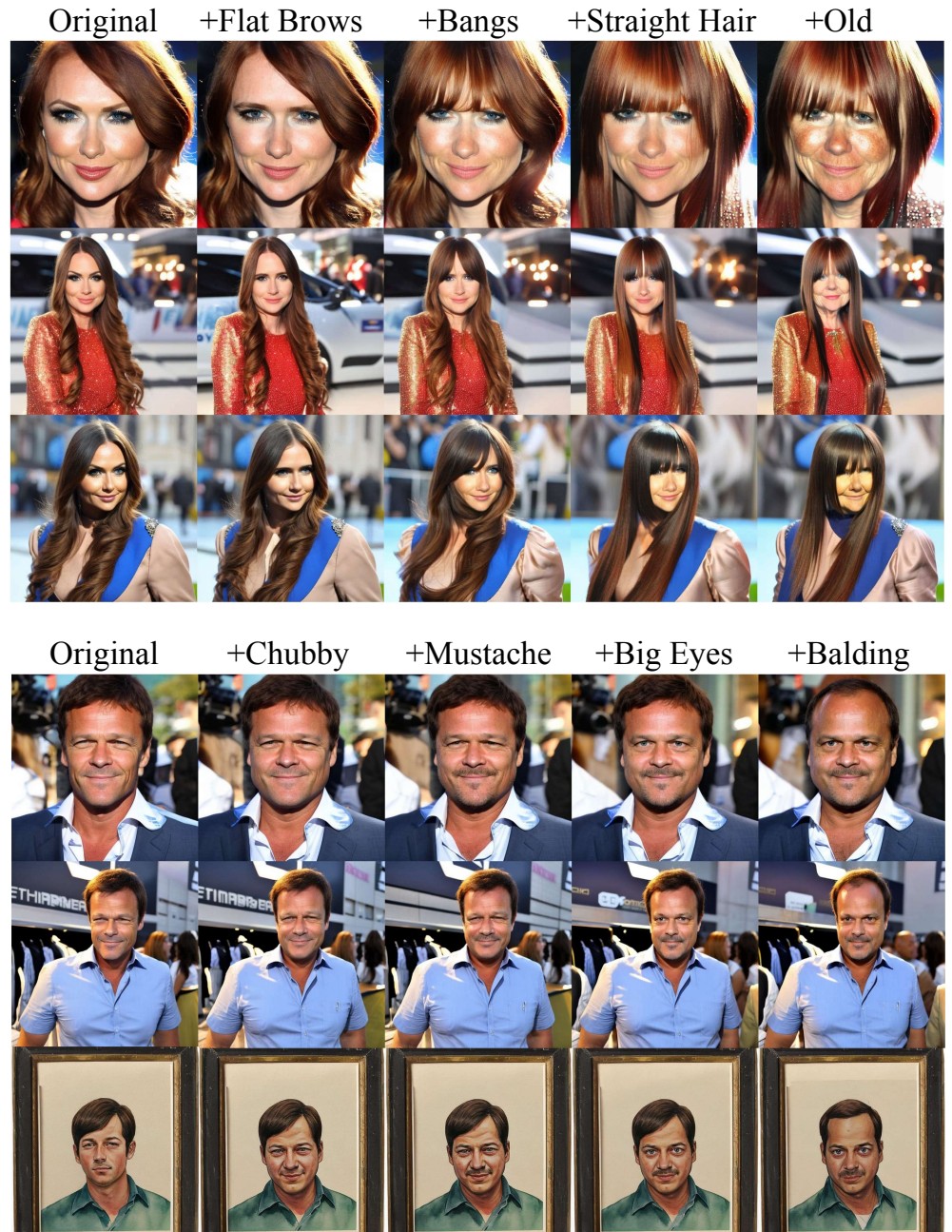

Figure 15: **Composing four different edits with minimal identity degradation**. These edits bind to the identity and persist in appearance across multiple generation seeds and prompts.

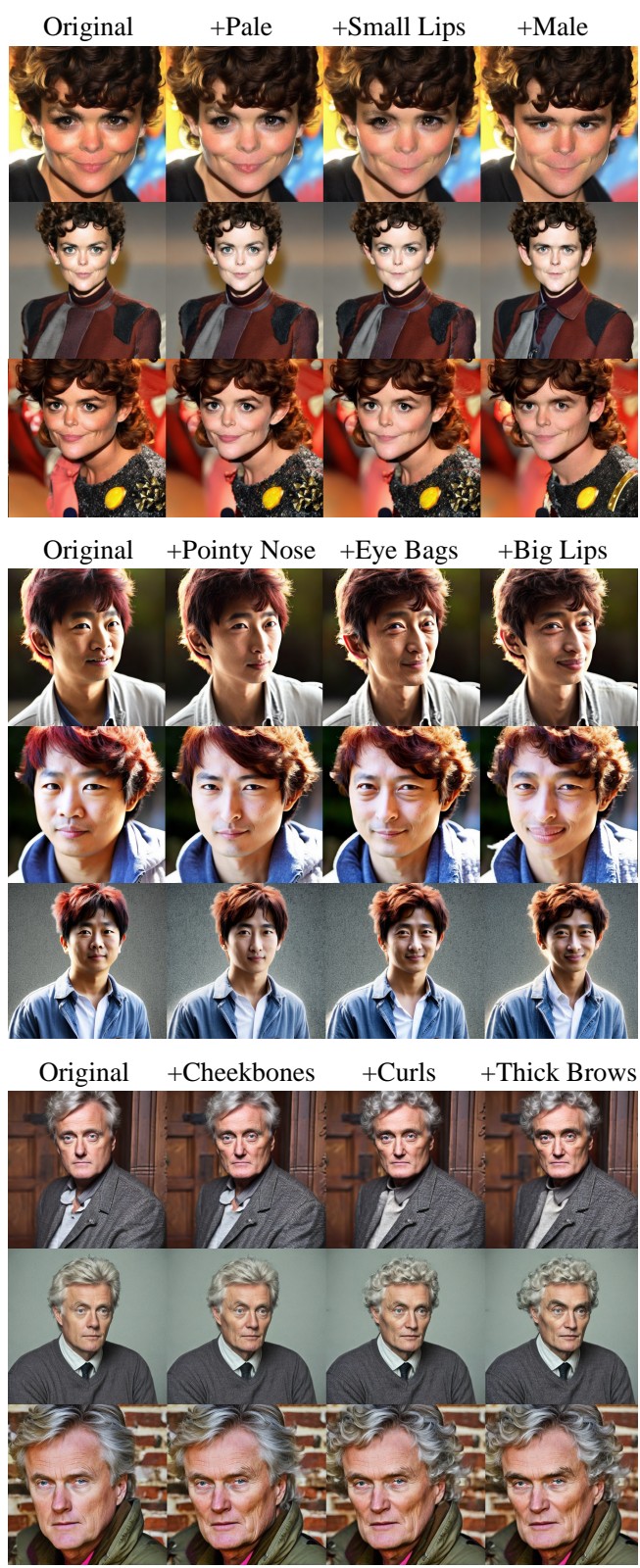

Figure 16: **Additional examples of composing multiple edits.** We provide more examples of semantic edits based on labels available from CelebA.

**User Study.** We present a two-alternative forced choice (2AFC) user study to evaluate the quality of identity edits in Tab. 3. Twenty-five users were given ten sets of images. Each set contained a randomly sampled original image of an identity, and then an image of that identity edited for an attribute using Concept Sliders [17], *w2w*, and text prompting. Users were then asked to choose between alternate pairs based on three criteria: identity preservation, alignment with the desired edit, and disentanglement. Our results in Tab. 3 show that users have a strong preference towards *w2w* edits. User instructions and an example question from the study are provided in Figs. 17, 18.

Table 3: **User study on identity editing.**

| Method | Win Rate (%) ↑ |
|--------|----------------|
| Sliders | 28.4 |
| *w2w* | **71.6** |
| Prompting | 12.8 |
| *w2w* | **87.2** |

## Comparing Edits

**Read the following instructions.**
In each page, there will be an original identity, followed by an attribute edited for that identity using three different methods. You will be asked 3 different questions. Each question will ask you to choose between two of the edit methods. The top of each section will provide the edit description. Choose which image **satisfies the most of these three criteria:**
**1. Semantic Alignment.** The image aligns with a textual description for the edit (e.g. "chubby" should mean that the edited identity now looks chubby).
**2. Identity Preservation.** The image preserves the original identity. In other words, the edited identity still looks recognizable compared to the original. Some things to look out for are: eye shape, nose shape, face shape (e.g., circular or oval-like), skin tone, eyebrow shape, hair color.
**3. Disentanglement.** The edit minimally interferes with the rest of the image (i.e. disentangled and does not affect clothes, background, etc.)

Figure 17: **Instructions provided to users for the identity editing user study.**

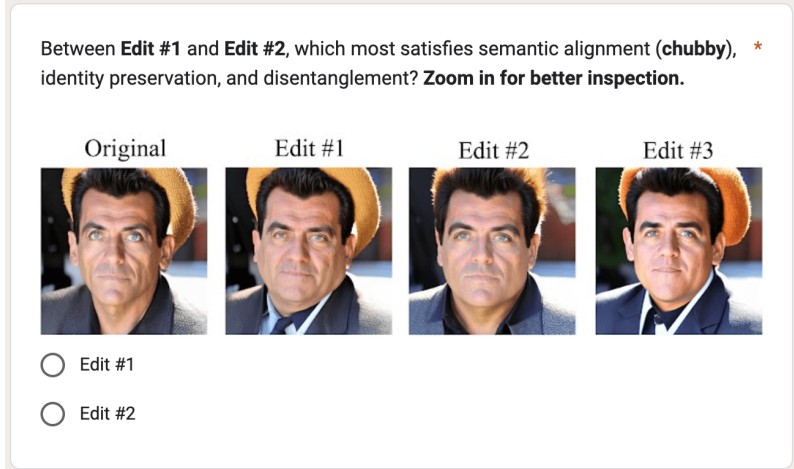

Between **Edit #1** and **Edit #2**, which most satisfies semantic alignment (**chubby**),   * identity preservation, and disentanglement? **Zoom in for better inspection.**

| Original | Edit #1 | Edit #2 | Edit #3 |

○ Edit #1

○ Edit #2

Figure 18: **Example question from the identity editing user study.**

## C  Inversion

We present additional details on *w2w* inversion and comparisons against training Dreambooth LoRA on a single image vs. multiple images.

**Implementation Details:** To conduct *w2w* inversion, we train on a single image following the objective from eq. 3. We qualitatively find that optimizing 10,000 principal component coefficients balances identity preservation with editability. This is discussed in Appendix F. We optimize for 400 epochs, using Adam [30] with learning rate 0.1, $\beta_1 = 0.9$, $\beta_2 = 0.999$ and with weight decay factor $1e$-10. For conducting Dreambooth fine-tuning, we follow the implementation from Hugging Face [3] using LoRA with rank 1. To create a dataset of multiple images for an identity, we follow the procedure from Sec. 4.4.

***w2w* inversion is more efficient than previous methods.** Inversion into *w2w* space results in a significant speedup in optimization as seen in Tab. 4, where we measure the training time on a single NVIDIA A100 GPU. Standard Dreambooth fine-tuning operates on the full weight space and incorporates an additional prior preservation loss which typically requires hundreds of prior images. In contrast, we only optimize a standard denoising objective on a single image within a low-dimensional weight subspace. Despite operating with lower dimensionality, *w2w* inversion performs closely to standard Dreambooth fine-tuning on multiple images with LoRA.

Table 4: **Inversion into *w2w* space balances identity preservation and efficiency.**

| Method | Single-Image | # Param | Opt. Time (s) | Identity Fidelity ↑ |
|---|---|---|---|---|
| DB-LoRA | × | 99,648 | 220 | **0.69** $\pm$ 0.01 |
| DB-LoRA | ✓ | 99,648 | 200 | 0.43 $\pm$ 0.03 |
| *w2w* Inversion | ✓ | 10,000 | 55 | 0.64 $\pm$ 0.01 |

**Qualitative Inversion Comparison.** In Figs. 19 and 20, we present qualitative comparisons of *w2w* inversion against Dreambooth trained with multiple images and a single image. Although *mult-image* Dreambooth slightly outperforms *w2w* inversion in identity preservations, its samples tend to lack realism compared to *w2w* inversion. We hypothesize that this may be due to using generated images for prior preservation and training on synthetic identity images. Dreambooth trained on a single image either generates an artifacted version of the original image or random identities. Notice how inversion into *w2w* space is even able to capture key characteristics of the child although babies are nearly to completely absent in the identites based on CelebA used to fine-tune our dataset of models.

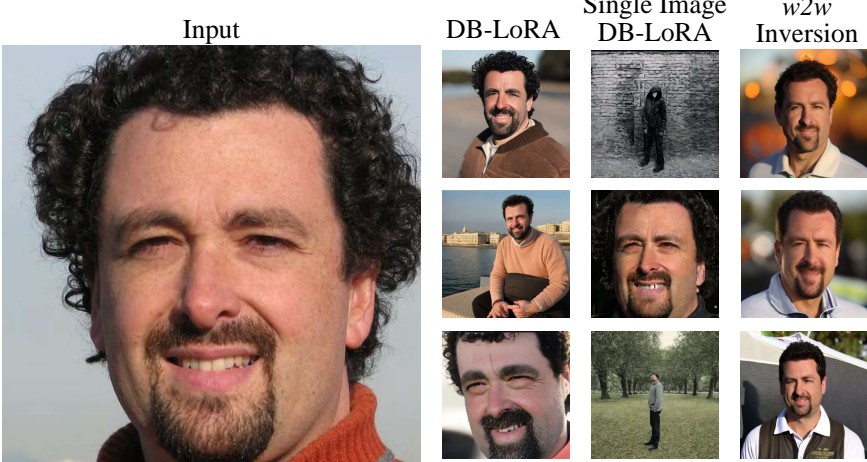

Figure 19: **Inversion into *w2w* space preserves identity and realism.** We compare against Dreambooth fine-tuning with LoRA on multiple images and a single image.

---

[3] https://github.com/huggingface/peft

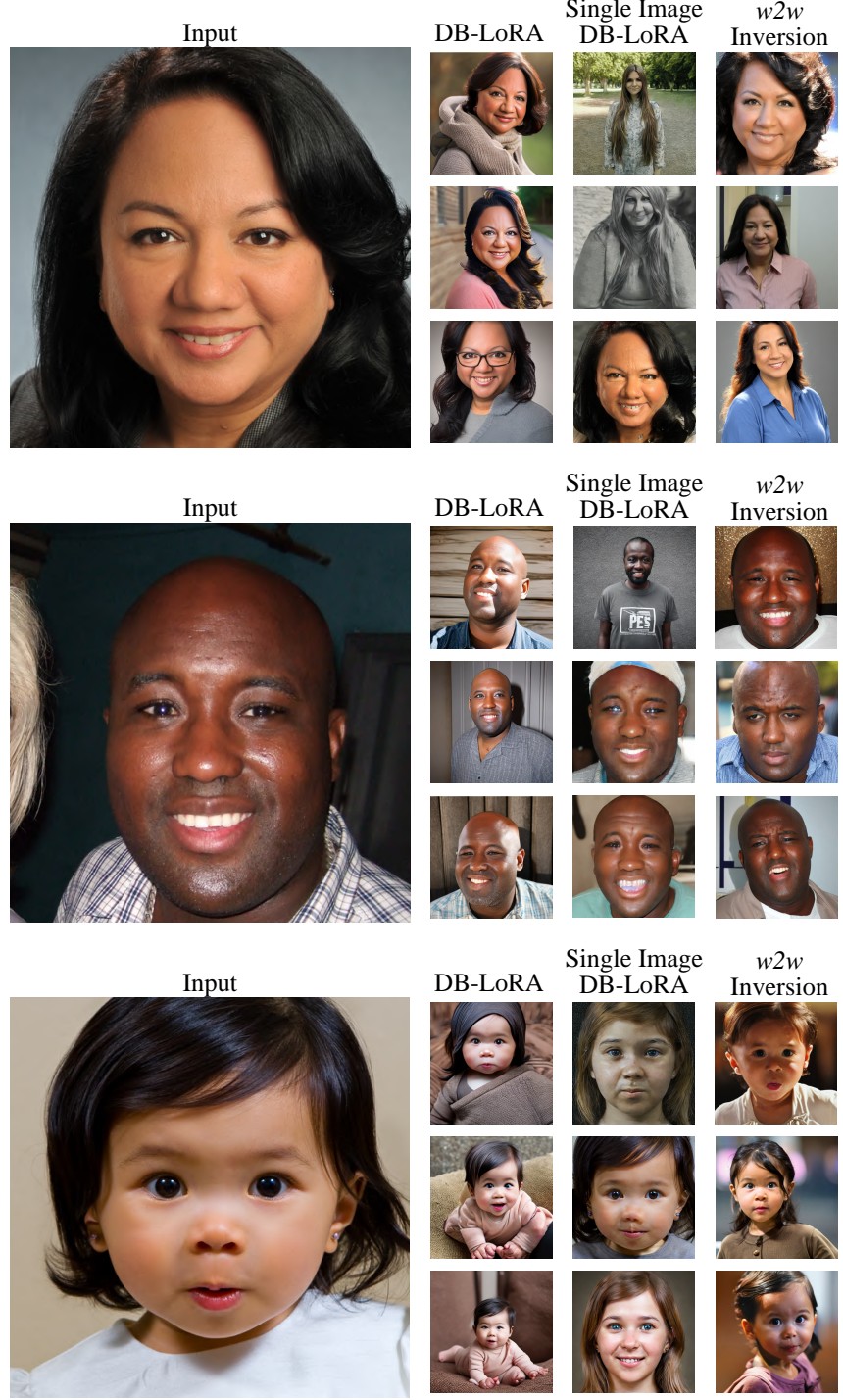

Figure 20: **Inversion into *w2w* space preserves identity and realism (cont.).**

**Comparison against Single Image Personalization Methods.** We compare *w2w* inversion to single shot personalization methods Celeb-Basis [73] and IP-Adapter FaceID[4] [72], following the same evaluation protocol from Sec. 4.4. These quantitative results are presented in Tab. 5. While Celeb-Basis optimizes in the input text embedding space for a given face image, IP-Adapter trains light-weight adapters [39, 75] to condition generation on any input face image.

We further conducted a two-alternative forced choice (2AFC) user study on the perceptual quality of *w2w* identity preservation. Twenty users were given ten sets of images. Each set contained a randomly sampled original image of the identity, and then three random images generated using IP-Adapter FaceID, Celeb-Basis, and *w2w* with the same random prompt. Users were then asked to choose between alternate pairs based on three criteria: identity preservation, prompt alignment, and diversity of generated images. Our results in Tab. 6 show that users found generations from *w2w* models capture identity better while also generating more diverse images that better align with the prompts.

Across both these metrics, *w2w* performs stronger than Celeb-Basis and IP-Adapter FaceID. Our results indicate that operating in our weight subspace is highly expressive and flexible as it is able to faithfully capture nuanced identity without overfitting to the input image. For instance, in Fig. 21, *w2w* inversion enables diverse generation with various poses, facial expressions, and clothing while maintaining identity.

Table 5: **Single-shot personalization comparison.**

| Method | ID Score ↑ |
|---|---|
| Celeb-Basis | $0.60 \pm 0.02$ |
| IP-Adapter FaceID | $0.62 \pm 0.02$ |
| *w2w* | $\mathbf{0.64 \pm 0.01}$ |

Table 6: **User study on identity inversion.**

| Method | Win Rate (%) ↑ |
|---|---|
| Celeb-Basis | 13.4 |
| *w2w* | **86.6** |
| IP-Adapter FaceID | 22.8 |
| *w2w* | **77.2** |

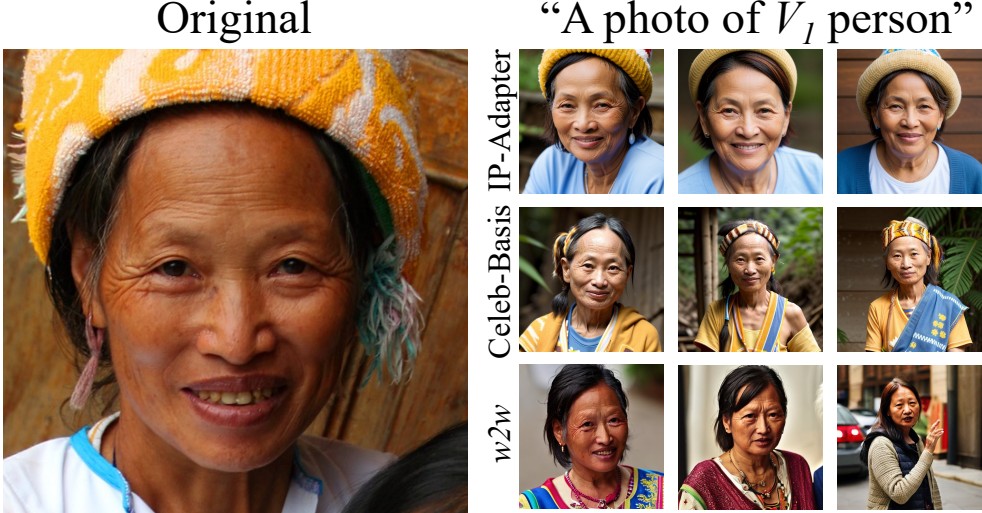

Figure 21: **Qualitative comparison of single-shot personalization methods.**

---

[4]https://huggingface.co/h94/IP-Adapter-FaceID

Below in Figs. 22 and 23, we provide the instructions provided to users for the identity inversion user study in addition to an example question.

## Comparing Identities

**Read the following instructions.**
In each page, there will be an original identity, followed by three sets of images trying to mimic that identity generated with a prompt. You will be asked 3 different questions. Each question will ask you to choose between two of the methods. The top of each section will provide the prompt used to generate the images. Choose which set of images **satisfies the most of these three criteria to a reasonable degree:**

**1. Identity Preservation.** The images preserve the original identity. In other words, the edited identity still looks recognizable compared to the original. Some things to look out for are: eye shape, nose shape, face shape (e.g., circular or oval-like), skin tone, eyebrow shape, hair color. **Try not to be biased by image quality, alignment of the pose, or facial expression.**

**2. Prompt Alignment.** The set of images should satisfy the prompt used to generate that set of images (e.g., if the prompt is "playing tennis outdoors," the set of images should satisfy that).

**3. Image Diversity.** The set of images should not just copy the original image. There should be a variety of poses, backgrounds, clothes, and facial expressions (if the prompt does not include those).

Figure 22: **Instructions provided to users for the identity inversion user study.**

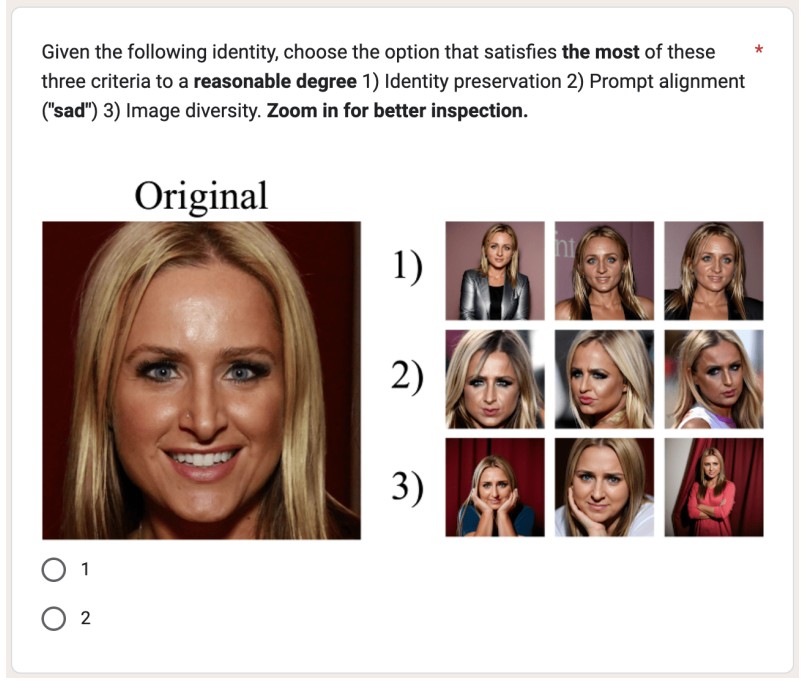

Figure 23: **Example question from the identity inversion user study.**

# D    Out of Distribution Projection

Additional examples of out-of-distribution projections are displayed in Fig. 24. A diverse array of styles and subjects (e.g. paintings, sketches, non-humans) can be distilled into a model in *w2w* space. After embedding an identity into this space, the model still retains the compositionality and rich priors of a standard personalized model. For instance, we can generate images using prompts such as "*[v]* person writing at a desk" (top example), "*[v]* person with a dog" (middle example), or "a painting of *[v]* person painting on a canvas" (bottom example).

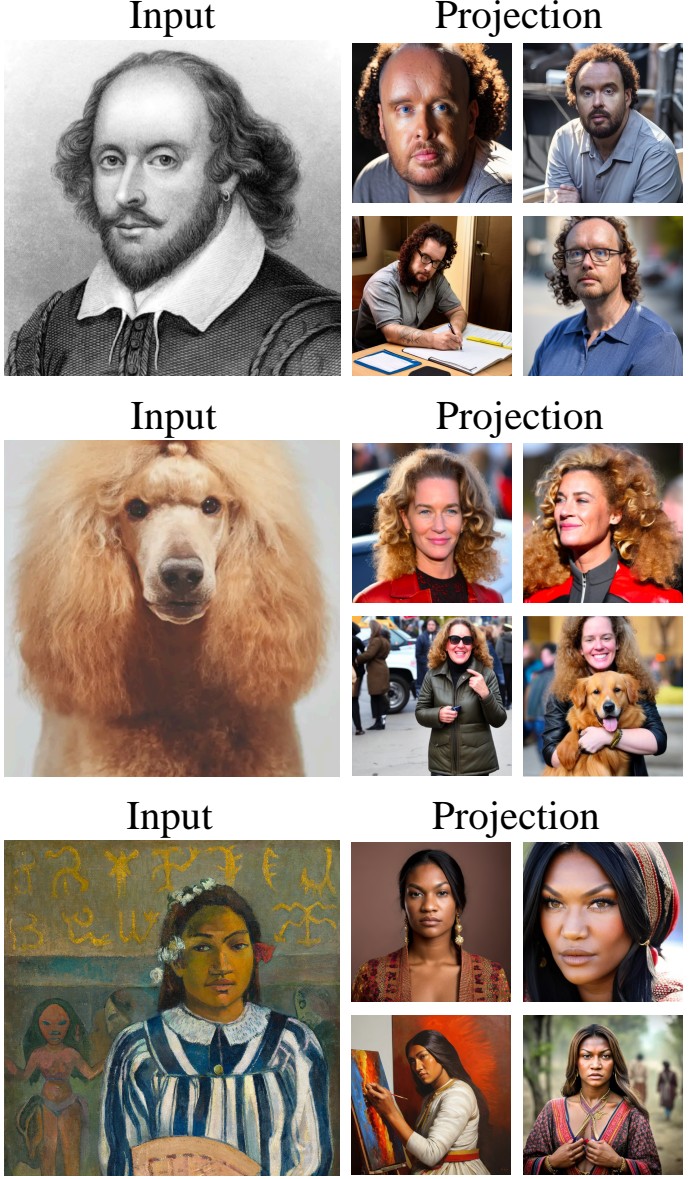

Figure 24: **Projection into *w2w* space generalizes to a variety of inputs.** A range of styles and entities can be inverted into a realistic identity in this space. Once a model is obtained, it can be prompted to generate the identity in a variety of contexts.

# E  Identity Datasets

In Fig. 25, we present examples of synthetic identity datasets used to conduct our Dreambooth fine-tuning as discussed in Sec 4. Each dataset is a set of ten images generated with [67] conditioned on single CelebA [36] images associated with binary attribute labels. Note that we only display a subset of images per identity in the figure. The same identity can occur multiple times in different images in CelebA but have different appearances. So, creating these synthetic datasets reduces intra-dataset diversity and creates a more consistent appearance for each subject. For instance, the first two rows in the figure are the same identity, but look drastically different. So we instead treat them as different identities associated with a different set of images.

For evaluating identity edits from Sec. 4.3, we hold out 100 identities, which results in leaving out ∼1000 models since multiple models may encode different instances of the same identity. For instance, if we left out the model encoding the identity in the first row of Fig. 25 for evaluation, the model encoding the second row identity would also be left out since it encodes the same identity but a different instance.

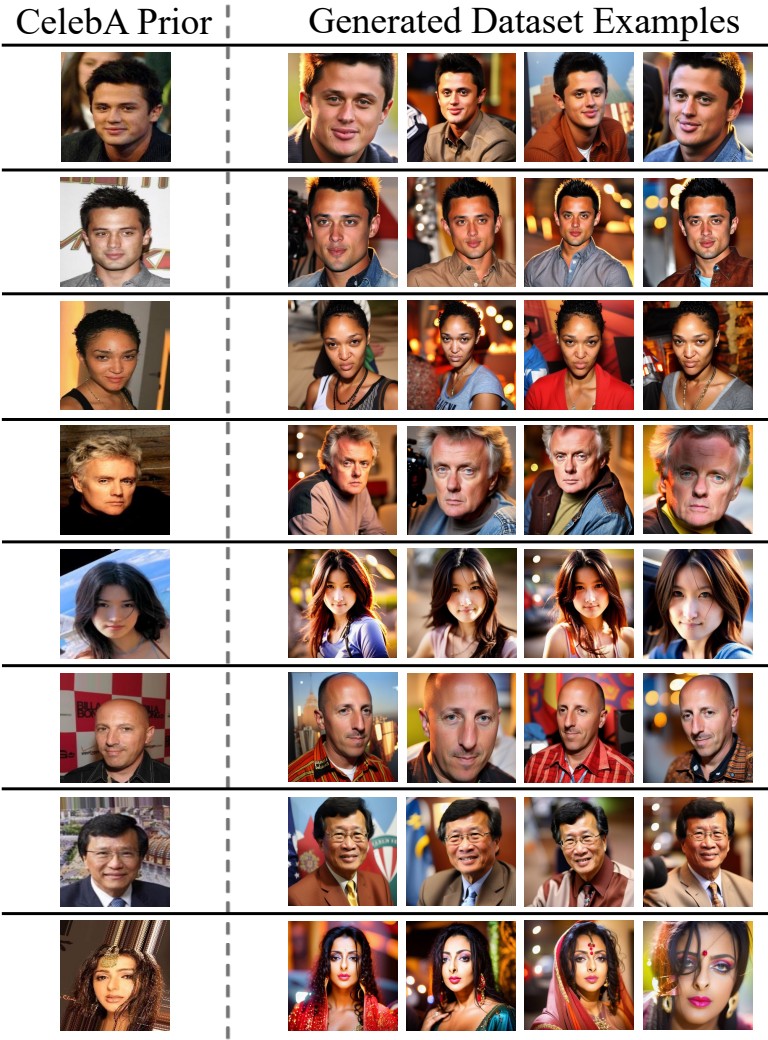

Figure 25: **Fine-tuning on synthetic examples allows Dreambooth fine-tuning to distill a consistent identity.** The left column shows a CelebA image used to condition generation of a set of identity-consistent images in the right column associated with that identity using [67]. The consistent appearance of the identity enables a more consistent identity encoding.

# F    Principal Component Basis

In this section, we analyze various properties of the Principal Component (PC) basis used to define *w2w* Space. We investigate the distribution of PC coefficients and the effect of the number of PCs on identity editing and inversion.

**Distribution of PC Coefficients.** We plot the histogram of the coefficient values for the first three Principal Components in Fig. 26. They appear roughly Gaussian. Next, we rescale the coefficients for these three components to unit variance for visualization purposes. We then plot the pairwise joint distributions for them in Fig. 27. The circular shapes indicates roughly diagonal covariances. Although the joint over other combinations of Principal Components may exhibit different properties, these results motivate us to model the PCs as independent Gaussians, leading to the *w2w* sampling strategy from Sec. 3.2.

**Number of Principal Components for Identity Editing** We empirically observe that training classifiers based on the 1000 dimensional PC representations (first 1000 PCs) of the model weights results in the most semantically aligned and disentangled edits directions. We visualize a comparison for the "goatee" direction in Fig. 28. After finding the direction, we calculate the maximum projection component onto the edit direction among the training set of model weights. This determines the edit strength. As seen in the figure, restricting to the first 100 Principal Components may be too coarse to achieve the fine-grained edit, instead relying on spurious cues such as skin color. Training with the first 10,000 Principal Components suffers from the curse of dimensionality and the discovered direction may edit other concepts such as eye color or clothes. Finding the direction using the first 1000 Principal Components achieves the desired edit with minimal entanglement with other concepts.

**Number of Principal Components for Identity Inversion** We qualitatively observe that inversion using the first 10,000 Principal Components balances identity preservation while not overfitting to the source image. We visualize a comparison in Fig. 29, where each column has a fixed seed and prompt. Optimizing with the first 1000 PCs underfits the identity and does not generate a consistent identity. Inversion with the first 20,000 Principal Components overfits to the source image of a face shot, which results in artifacted face images despite different generation seeds and prompts. Optimizing with the first 10,000 Principal Components enjoys the benefits of a lower dimensional representation than the original LoRA parameter space (∼100,000 trainable parameters), while still preserving identity and compositionality. This is supported quantitatively by Fig. 30, which shows the average ID score for 100 inverted FFHQ identities optimized over a varying number of principal components.



Figure 26: **Histogram of principal component coefficients.** The first three principal component coefficients appear approximately Gaussian.



Figure 27: **Pairwise joint histogram of principal component coefficients.** We rescale the first three principal component coefficients and plot the pairwise joint distributions for visualization purposes. Given that the marginals are roughly Gaussian, the circular appearance of the joint suggests pairwise independence for the first three components.

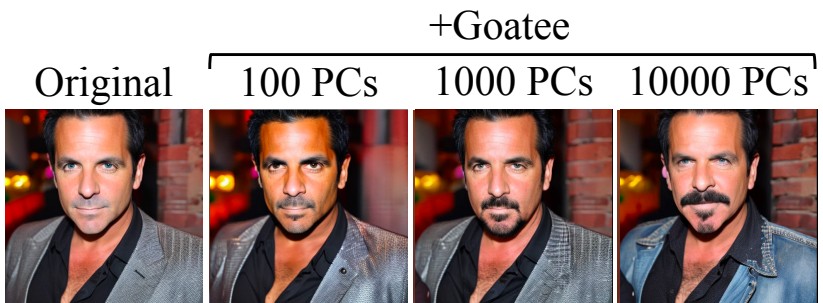

Figure 28: **Edit results with varying number of Principal Components.** Training classifiers to find semantic weight space directions with the first 1000 Principal Components achieves the most semantically aligned and disentangled results.

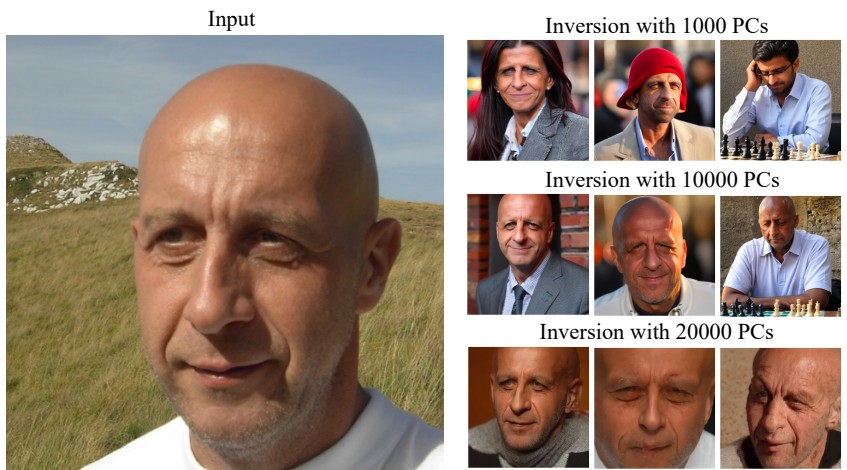

Figure 29: **Identity inversion results with varying number of principal components.** We optimize the coefficients for the first 1000, 10, 000, and 20, 000 Principal Component. Each column indicates a fixed generation seed and prompt. Inversion with the first 10, 000 components balances parameter efficiency, realism, and identity preservation without overfitting to the single image.

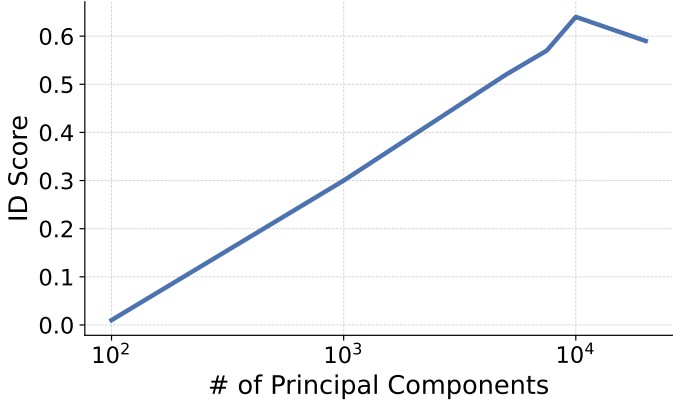

Figure 30: **Identity preservation vs. number of principal components used for *w2w* inversion.** We optimize the coefficients for the first $N$ principal components up to 20,000 and measure the average ID score for 100 inverted FFHQ identities.

**Visualizing Principal Components.** We provide a visualization of traversals along a set of principal components in Fig. 31. The principal components change attributes of the identity, although various semantic attributes are entangled. For instance, the first PC appears to change age, hair color, and hair style. The second PC appears to change gender and skin complexion. The third PC seems to change age, skin complexion, and facial hair. This motivates our use of linear classifiers to find separating hyperplanes in weight space and disentangle these attributes.

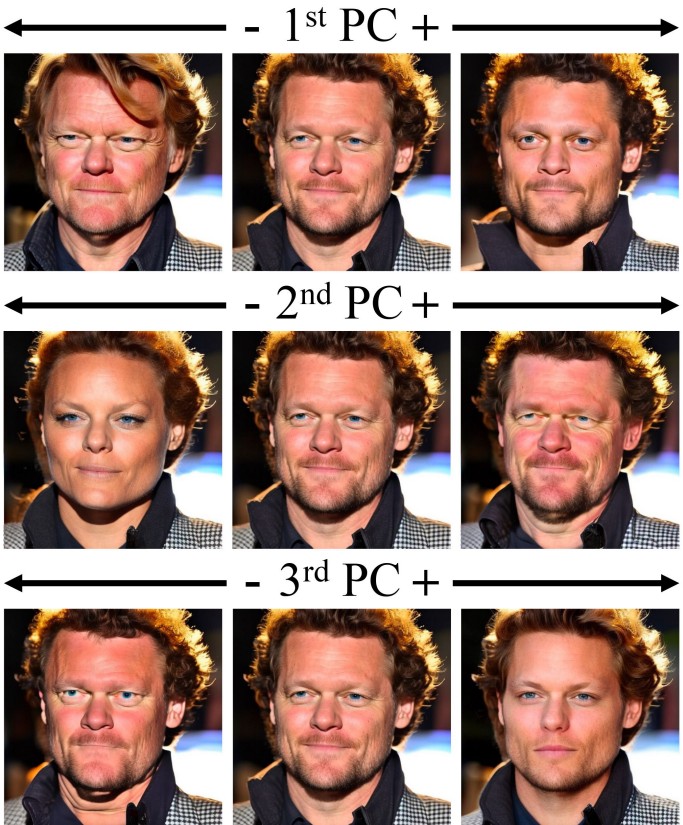

Figure 31: **Traversal along the first three principal components in *w2w* space.** The directions encode various entangled identity attributes such as age, gender, and facial hair.

## G  *weights2weights* for Other Visual Concepts

We find that similar subspaces can be created for other visual concepts beyond human identites. For instance, we apply the *weights2weights* framework to create two subspaces for models encoding different dog breeds and models encoding car types. We present examples of editing these models in Fig. 32. This suggest the generality of *weights2weights* and linear subspaces within diffusion model weights.

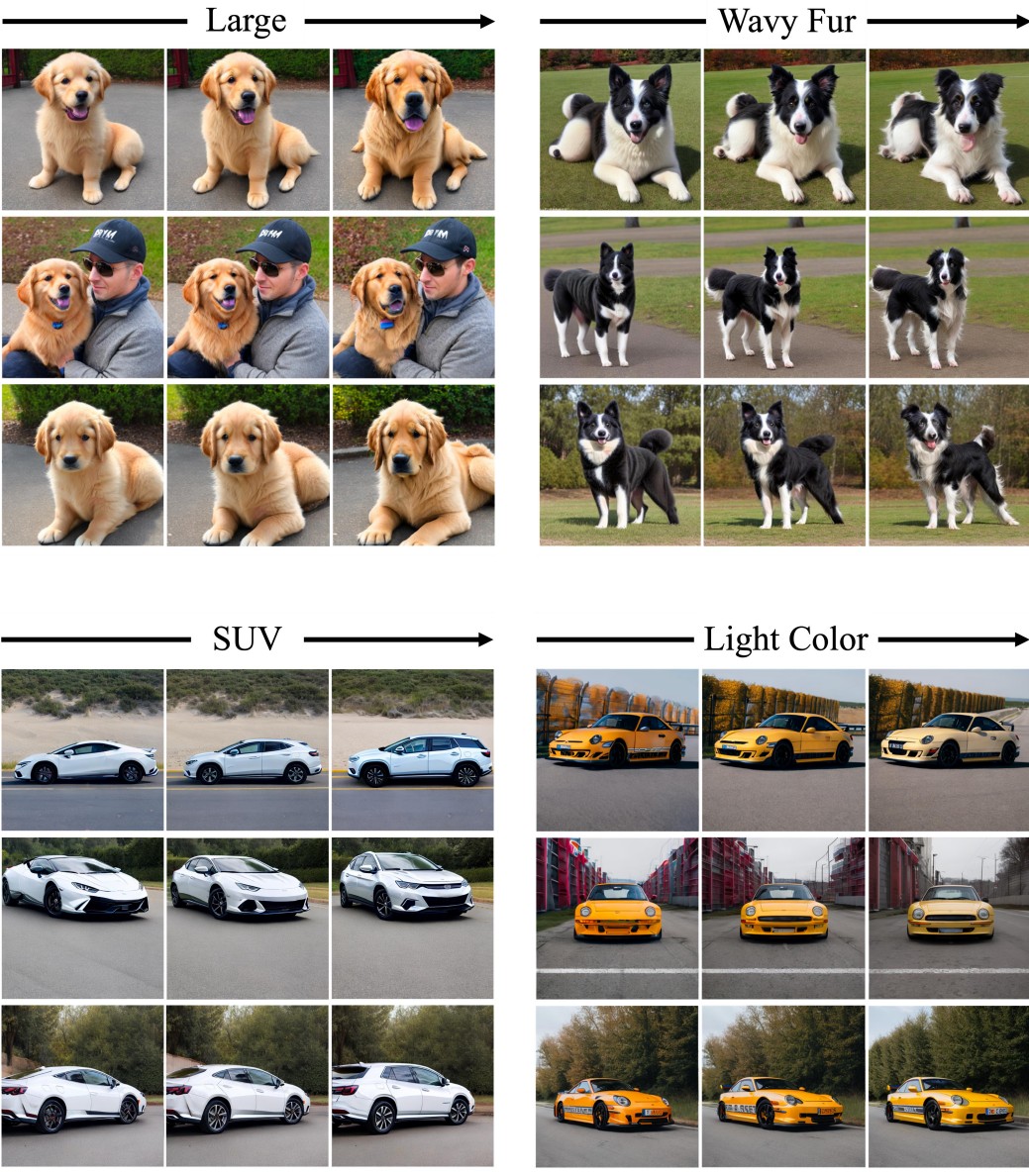

Figure 32: **Applying *weights2weights* edits on dog-encoding models and car-encoding models.** We create two datasets of model weights and apply PCA to define two separate weight subspaces. We then train linear classifiers to find semantic edit directions.

# H   Multi-Concept Merging

Multiple models living in *w2w* space cannot be merged since they live in the same weight subspace. So, merging will lead to interpolation of the identities. However, *w2w* models can be merged with models lying approximately orthogonal to the subspace. This merging can be done adding the weights. We present an example of merging a model living in *w2w* space with a model fine-tuned to encode "Pixar" style in Fig. 33.

A photo of $V_1$ person

A photo of $V_1$ person in $V_2$ style

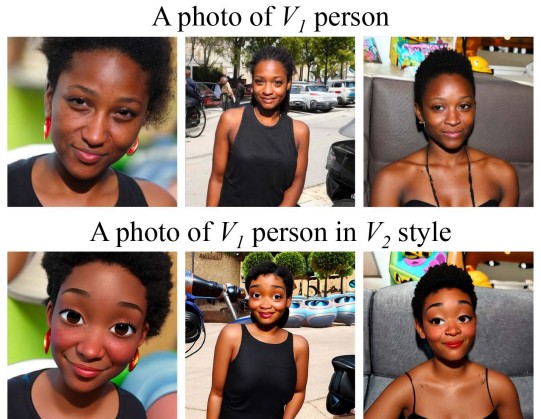

Figure 33: **Merging *w2w* models with non-identity models.** Here, another model is fine-tuned to map $V_2$ to "Pixar" style. The two models are merged with simple addition.

# I   Timestep Analysis

Edits in *w2w* space correspond to identity edits with minimal interference with other visual concepts. Although not a focus, image editing is achieved as a byproduct. For further context preservation, edits in *w2w* Space can be integrated with delayed injection [7, 17, 35, 71], where after $T$ timesteps, the edited weights are used instead of the original ones. We visualize this in Fig. 34. Larger $T$ in the range $[700, 1000]$ are helpful for more global attribute changes, while smaller $[400, 700]$ can be used for more fine-grained edits. However, by decreasing the timestep $T$, the strength of the edit is lost in favor of better context preservation. For instance, the dog's face is better preserved in the second row at $T = 600$, although the man is not as chubby compared to other $T$.

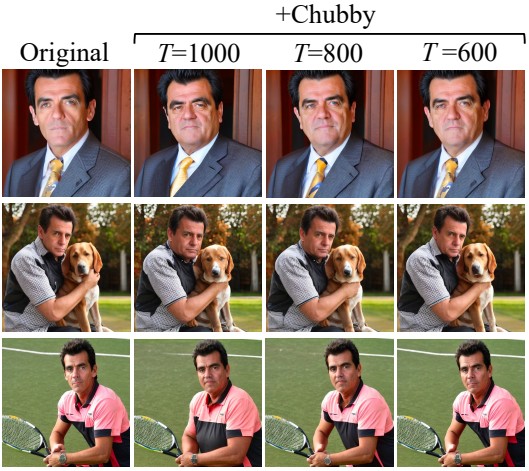

Figure 34: **Injecting edited weights at varying timesteps.** Using the edited weights at a smaller timestep $T$ better preserves context at the expense of edit strength and fidelity.

