# OpenReview forum: "Interpreting the Weight Space of Customized Diffusion Models"
_NeurIPS.cc/2024/Conference — NeurIPS 2024 poster_

### Official Review · Reviewer_215A · 2024-07-02

**Soundness:** 3
**Presentation:** 3
**Contribution:** 2
**Rating:** 6
**Confidence:** 5

**Summary:**

The paper proposes to learn a manifold over customized diffusion model weights as a subspace for interpretable downstream applications such as sampling, editing, and inversion.  At the technological level, the authors collect the weights of over 60,000 models as the dataset and fine-tune those weights via LoRA to constitute the desired editing space. Experiments are conducted on CelebA dataset.

**Strengths:**

- The paper is generally well organized and easy to follow, with reasonable motivation and intuitions.

- The idea of using weights as the dataset to constitute an editing space is interesting.

- Among the experimented application scenarios, while editing and inverting have been widely studied in the field (see Weaknesses), the exploration of the OOD projection seems more interesting to me.

**Weaknesses:**

- The paper lacks a comprehensive comparison with prior literature, resulting in inaccurate claims on “Alas, in multi-step generative models, like diffusion models, such a linear latent space is yet to be found”. In [a], it has been explicitly and theoretically proven such interpretable and linear latent space can be found in frozen DMs, without the need to fine-tune the model parameters, neither the DreamBooth tuning nor LoRA-based. Recent studies [2,3] have extended these findings for customization tasks and latent diffusion models. These methods have covered the scenarios including latent sampling, editing, and inverting, and thus should be discussed.

- From the technical point of view, the authors put much more emphasis on the empirical investigations on the interpretability of latent spaces in DMs, but neglect the dimension of diffusion time steps from a more fundamental perspective. It would strengthen the paper if the authors could elaborate on the impact of time steps on the construction of your w2w space because multiple existing works [a,b,d] suggest the time step of latent spaces in DMs could carry important information on the distribution learning process and thus influence the downstream tasks.

- From the experiments and evaluation perspective, it would be helpful to perform some user studies as subjective evaluations as many other generative application works do. In addition, these attribute directions and their properties have been demonstrated other than identity context, it may strengthen the paper if the authors could extend their application scenarios to a wider context.

---

[a] Boundary Guided Learning Free Semantic Control with Diffusion Models, NeurIPS 2023.

[b] Diffusion in Diffusion: Cyclic One-Way Diffusion for Text-Vision-Conditioned Generation, ICLR 2024.

[c]  Continuous, Subject-Specific Attribute Control in T2I Models by Identifying Semantic Directions. 2024

[d] Diffusionclip: Text-guided diffusion models for robust image manipulation. CVPR 2022.

**Questions:**

Please see my comments in the Weaknesses section.

I have mixed feelings about this paper. On the one hand, the paper risks over-claiming on some high-level motivations as pointed out in the Weaknesses section, lacking a clear positioning concerning several important related works. The methodological level design is also not entirely new and intriguing. On the other hand, I acknowledge that using model weights as datasets is interesting, and the efforts of “scaling up” to over 60.000 models seem to be one of the biggest contributions to me.

**Limitations:**

The limitations and broader societal impact are discussed in the main paper.

---

> ### Author Rebuttal · Authors · 2024-08-06
>
> **“...editing and inverting have been widely studied in the field (see Weaknesses)”** and **“These methods have covered the scenarios including latent sampling, editing, and inverting and thus should be discussed”** and **“The methodological level design is also not entirely new and intriguing.”**
>
> We agree that latent sampling, editing, and inverting has been studied extensively. However, we believe there might be a misunderstanding. We explore these applications on **model weights**, not input noise or latents. *weights2weights* space can be thought of as a **meta**-latent space, with  “applications analogous to those of a generative latent space – inversion, editing, and sampling – but **producing model weights rather than images**” (L316-318 in paper). We agree that the works you bring up are relevant, and we will include them in the related works. However, they focus on image manipulation, operating on the **image manifold**. As pointed out by reviewers rzPz and wMSF, we model and operate on the **“model weight manifold,”** manipulating the models themselves in order to produce new models. To the best of our knowledge, we believe this idea of a **meta**-latent space over model weights is novel. Furthermore, we add to this contribution by finding semantic weight subspaces with PCA and linear classifiers. Our analysis of the model weights shows that we can get valid identity-encoding diffusion models via linear interpolation among the weights of existing diffusion models spanning the subspace.
>
> We contrast sampling, editing, and inverting in the traditional sense with our idea of operating on model weights in Figure 2 of our paper. **1)** Sampling in the traditional sense will produce a latent that results in a single image. Sampling from *w2w* space produces a **new diffusion model** that can generate an infinite number of different images of the same identity. **2)** Editing an input such as a latent results in one edited image. In contrast, edits in *w2w* space result in a **new model** with edited weights such that attributes of the identity that this model generates are changed. Once the edit is conducted, we can generate new images from the model and the edited identity is consistent (e.g., now always has a beard). Throughout the paper, we show results of generated images before and after the model weight edits and observe that the generated images are largely unchanged except for the identities. This is to show that after editing the weights, the model is largely unaffected except its identity. **3)** Inversion into the input space in generative models reconstructs a single image. However, inversion into *w2w* space produces a **new model** based on the identity in the input. We can then indefinitely sample new images from the model which consistently generates that identity. Furthermore, with the OOD projection, we are distilling a realistic identity from the input image into a new model.
>
> **“... resulting in inaccurate claims on ‘Alas, in multi-step generative models, like diffusion models, such a linear latent space is yet to be found.’”**
>
> Thank you for pointing this out. We agree that this sentence is poorly phrased. The point we meant to get across is nicely summarized in paper [a] that you brought up: “In contrast to a single unified latent space in GANs [the multistep process of diffusion models] imposes extra difficulties for studying the latent spaces of DDMs.” Still, our motivation for studying the weight space and modeling its manifold remains. The fact that personalization approaches can insert concepts into the model weights without disrupting the prior of the model suggests that we can find such subspaces. This inspires us to develop the *weights2weights* as a *meta-latent space* controlling model weights. We will clarify this in the manuscript.
>
>
> **“The authors put much more emphasis on the empirical investigations on the interpretability of latent spaces in DMs, but neglect the dimension of diffusion time…”**
>
> We agree that analysis of diffusion timesteps is interesting with regards to the latent or noise spaces in DMs. However, we are analyzing the weights, which are fixed across timesteps. The identity and attribute subspaces we find are in the weights, and the applications such as inversion, sampling, and editing produce new weights parameterizing a new model which is fixed across timesteps.
>
>
> **“... it would be helpful to perform some user studies as subjective evaluations…”**
>
> Thank you for your suggestion! We conducted user studies for identity inversion in rebuttal PDF Table 2 and identity editing in rebuttal Table 3. We discuss the details of these in global responses 3 and 4. Overall, users have a strong preference towards *weights2weights* for identity inversion and editing compared to baselines.
>
>
>
> **“...other than identity context…it may strengthen the paper if the authors could extend their application scenarios to a wider context.”**
>
> We agree with your suggestion. Following your comment, we create a *weights2weights* weight subspace for dogs and  present the results in rebuttal PDF Figure 3. We conduct Dreambooth fine-tuning for 120 different dog breeds and train linear classifiers to define separating hyperplanes in weight space for different semantic attributes, such as “large” or “wavy fur.” In rebuttal Figure 3, we sample two models from this space and edit the weights to change attributes of the encoded breeds.
>
>
> **"...I acknowledge that using model weights as datasets is interesting…‘scaling up’... one of the biggest contributions to me.”**
>
> Thank you for acknowledging the scale of our weight space analysis! We will open source our code and release the weights for the community.

---

> > ### Comment · Reviewer_215A · 2024-08-11
> > **post rebuttal**
> >
> > I appreciate the author's efforts in preparing the rebuttal.
> >
> > While I understand that this work focuses on **parameter weights**, I remain unconvinced (with a high level of confidence) that this parameter space and image space with noises should be completely independent, as the optimization of those parameters is driven by objectives that are intrinsically conditioned on the images.
> >
> > Nevertheless, as I noted in my initial review, the work has its merits, and the authors have addressed at least some of my concerns in the rebuttal. In light of this, I am raising the score to 6.

---

> > > ### Author Response · Authors · 2024-08-11
> > >
> > > Thank you for your further comments. We are happy to hear that we sufficiently answered some of your concerns, and we appreciate you raising your score. We agree with you that the weight space and noise space are not completely independent. This can make for some interesting future work.

---

### Official Review · Reviewer_wMSF · 2024-07-08

**Soundness:** 4
**Presentation:** 4
**Contribution:** 3
**Rating:** 5
**Confidence:** 3

**Summary:**

The paper investigates the weight space spanned by a large collection of customized diffusion models, proposing a novel subspace termed weights2weights (w2w). The study populates this space with a dataset of over 60,000 models, each fine-tuned to encode a different person's visual identity. The paper demonstrates three main applications of this space: sampling novel identities, editing existing identities, and inverting a single image to reconstruct realistic identities, showing that these weight spaces can serve as a highly interpretable latent identity space.

**Strengths:**

1. The concept of modeling a manifold over customized diffusion model weights is novel and shows potential for broad applications in generative modeling.

2. The PCA and LoRA are used to construct the w2w space. Furthermore, the PCA can be used to edit the image attributes. The method is novel and effective.

3. The paper includes extensive experiments that demonstrate the effectiveness of the proposed w2w space in generating, editing, and inverting identities.

**Weaknesses:**

1. The construction of w2w space needs over 6000 weights and the number of the model can further improve the ID score.  The computing resources should be provided. And is there a contained solution to balance computing resources and performance?

2. The paper uses several evaluation metrics (e.g., identity preservation, LPIPS, CLIP score), some metrics like CLIP score can describe the quality of image generation, but might not fully evaluate the nuanced attributes related to human faces.

3. In the weights manifold, new models can be sampled to find directions that separate identity properties. However, are the first m principal components all identity attributes, and do they correspond one-to-one?

**Questions:**

Please refer to the weaknesses above.

**Limitations:**

The authors have discussed the limitations of this work.

---

> ### Author Rebuttal · Authors · 2024-08-06
>
> **“The construction of w2w spaces needs over [60,000 models]...The compute resources should be provided…”**
>
> Training a single identity LoRA with rank 1 for our dataset of models requires ~8GB VRAM, and takes 220 seconds on a single A100 GPU. Please refer to Appendix C for details.
>
> **“And is there a contained solution to balance computing resources and performance.”**
>
> Thanks for the great question. We ablate the number of models used to construct *w2w* space in Figure 9 of the paper, and find that a subspace spanned by 10,000 models retains most of the performance in ID preservation with a score of 0.53 versus 0.65. The score of 0.65 is achieved by using more than six times the number of models. We highlight that although creating this space requires a notable amount of compute, using the space for the variety of applications is lightweight. For instance, inverting an identity only requires one image and uses 1/10 the number of parameters (just 10000) and 1/4 the time (under 55 seconds) for typical Dreambooth LoRA finetuning (see Appendix C).
>
> **“The paper uses several evaluation metrics…nuanced attributes related to human faces”**
>
> We agree with your point on the limitations of the reported metrics, acknowledging it in L226-227: “...score can be noisy...too coarse to describe attributes as nuanced as those related to the human face.” We utilize these metrics since they are commonly adopted for evaluating edits and identity preservation. Following your point, we present user studies for identity inversion in Table 2 and identity editing in Table 3 of the rebuttal PDF. Overall, *w2w* is largely preferred over baselines. We discuss this further in Global Responses 3 and 4.
>
> **“...are the first m principal components identity attributes…?”**
>
> That is an interesting question! Thank you for bringing it up. We provide a visualization of traversals along a set of principal components in Figure 4 of the rebuttal. The principal components change attributes of the identity, although various semantic attributes are entangled. For instance, the first PC seems to be changing age, hair color, and hair style. The second PC appears to change gender and skin complexion. The third PC seems to change age, skin complexion, and facial hair. This motivates our use of linear classifiers to find separating hyperplanes in weight space and disentangle these attributes.

---

### Official Review · Reviewer_rzPz · 2024-07-11

**Soundness:** 3
**Presentation:** 3
**Contribution:** 3
**Rating:** 7
**Confidence:** 5

**Summary:**

This paper constructs a dataset that contains LoRA models trained on images of 60,000 different people. A weight manifold is determined based on the parameters of these 60,000 models using PCA. Sampling, editing, and inversion can be performed on this manifold, and the rationality of this manifold is demonstrated.

**Strengths:**

1. This paper constructs a dataset that contains LoRA models trained on images of 60,000 different people.
2. This paper demonstrates the existence of a LoRA model manifold that encapsulates identity information and enables the generation of new identities by sampling from this manifold.
3. The paper also presents methods of editing and inversion, expanding the scope of manifold applications.

**Weaknesses:**

1. There are some recent works present the way of generating identity preserved images given only one reference image. It is better to provide the comparison between this paper and these works.
2. LoRA training often affects the performance of the base model to some extent, such as small generation variance. More results to examine this effect will be better.

**Questions:**

Refer to the Weaknesses 1 and 2.

**Limitations:**

Yes, limitations are discussed in this paper.

---

> ### Author Rebuttal · Authors · 2024-08-06
>
> **“...some recent works present the way of generating identity…given only one reference image…provide the comparison between this paper and these works.”**
>
> Thank you for your suggestion. In the rebuttal PDF Table 1, we compare against Celeb-Basis [1] and IP-Adapter FaceID [2], following the same evaluation protocol used for Table 2 in the main paper. We further conducted a user study comparing with [1] and [2], which is presented in Table 2 of the rebuttal PDF. We show qualitative comparisons against these two methods in rebuttal PDF Figure 6. Details are provided in Global Responses 2 and 3. Across all these metrics, *w2w* performs stronger than Celeb-Basis and IP-Adapter FaceID. Our results indicate that operating in the *weights2weights* weight subspace is highly expressive as it is able to faithfully capture identity without overfitting to the input image. This enables diverse generation of the identity.
>
>
> **“LoRA training often affects the performance of the base model to some extent…”**
>
> We are aware of this phenomenon known as “drift” as introduced in Dreambooth [4]. We agree with your concern of LoRA training affecting the base model, but we believe this is not necessarily an issue of *weights2weights* itself, but fine-tuning in general. Section 4.3 in Dreambooth [4] and Section 4.4 in Custom Diffusion [5] already run experiments indicating that the use of prior-preservation regularization limits the undesirable change of the base model. That is why we use the prior preservation loss introduced by Dreambooth to avoid losing the priors of the base model. Furthermore, the *w2w* models inherit the rich priors of the base model as shown by composition of the identities in novel contexts such as posing with a famous figure in Figures 6, 7 of the paper. These identities can also be converted into new styles with prompts like “statue” or “painting”  as seen in Figures 7, 8 in thr paper. More examples demonstrate this compositionality throughout the qualitative examples in the appendix.

---

### Official Review · Reviewer_aBHC · 2024-07-12

**Soundness:** 3
**Presentation:** 3
**Contribution:** 3
**Rating:** 5
**Confidence:** 4

**Summary:**

This paper explores the latent space of weights in customized diffusion models, introducing the weights2weights (w2w) space, a subspace encoding different human identities. By fine-tuning over 65,000 models, each representing a distinct human identity, the authors model this weight space using low-rank adaptation (LoRA) and principal component analysis (PCA). The key contributions include demonstrating three applications of the w2w space: Sampling: Generating novel identities by sampling from the weight space; Editing: Performing semantic edits, such as adding a beard, by traversing linear directions in the weight space; Inversion: Reconstructing a realistic identity from a single image, even if it is out-of-distribution. The paper validates the w2w space’s expressiveness through quantitative evaluations and qualitative observations.

**Strengths:**

1) The w2w space is interesting because it shifts the focus from traditional pixel or feature spaces to the model’s weight space, allowing for new forms of manipulation and analysis.

2) The paper is well-organized and clearly written, with detailed explanations of the proposed methods and their theoretical underpinnings.

**Weaknesses:**

1) The overall idea of this paper is more similar to [1], the significant difference I see so far is that [1] does PCA on token embedding and w2w does PCA on Lora in Unet. I suggest the authors could discuss more about the differences between this paper and [1]. Also, I have some methodological doubts:

A. does w2w have a similar phenomenon to mean person in [1]? (see Figure 2 in [1])

B. Have the authors studied the effect of PCA on the generation of results? (see section 4.3 in [1])

[1] Inserting Anybody in Diffusion Models via Celeb Basis

2) While this paper provides new ideas and insights into personalized generation models, I am concerned about his real-world application, what are the advantages of w2w compared to existing personalized generation efforts such as IP-adapter, InstabntID, custom diffusion, Mix-of-show, etc.? Faster generation or smaller training parameters? The experiments also don't give a comparison with existing personalization generation efforts, however, I think this is important to note

**Questions:**

1) I'm curious if style information can be decoupled via w2w space in addition to attributes. I would suggest that the authors could add the results of style editing to their paper.

2) How does w2w perform on multi-concept generation?

**Limitations:**

The paper discussed the limitations of this work.

---

> ### Author Rebuttal · Authors · 2024-08-06
>
> **“The overall idea of this paper is more similar to [1]...I suggest the authors could discuss more about the differences between this paper and [1].”**
>
> Although these two papers share the use of PCA and the application to personalization, there are fundamental differences. Many works have found subspaces in the *input space*, such as text embeddings [1], to produce meaningful linear interpolations. But as you point out, we use PCA to discover a subspace within the *weights themselves*, suggesting that weights can also be interpretable spaces. Our analysis of the model weights shows that we can get valid identity-encoding diffusion models via linear interpolation among the weights of existing diffusion models spanning the subspace. For instance, our *w2w* inversion essentially finds the best linear combination of existing diffusion model weights to define a **new customized diffusion model**, which, in turn, can generate an unlimited number of identity-consistent images of that novel person.
>
> Furthermore, beyond personalization, which is the main focus of [1], we demonstrate two additional applications of this subspace. We also develop methods for sampling and editing identities, which “expands the scope of manifold applications” (rzPz). We show how to **sample new models** each of which **encodes a novel identity** (Sections 3.2, 4.2), and **edit the models** (Sections 3.3, 4.3), which as a result, **edits the identities** encoded in them. As such, *weights2weights* space can be thought of as a **meta**-latent space as its applications are “analogous to those of a generative latent space – inversion, editing, and sampling – but producing model weights rather than images” (L316-318 in paper). We will make sure to clearly highlight these differences with Celeb-Basis in the related works section of the paper, where we briefly introduced Celeb-Basis.
>
>
> **“...similar phenomenon to mean person in [1]?”**
>
> *w2w* has a mean person, as shown in rebuttal Figure 5. These images are generated by a “mean model” using the averaged weights of models spanning *w2w* space.
>
> **“Have the authors studied the effect of PCA on the generation of results?”**
>
> Thank you for the suggestion. We present an ablation on the number of principal components used during *w2w* inversion in rebuttal PDF Figure 1. We vary the number of principal components used during identity inversion, and measure the average ID score (higher is better) over 100 inverted FFHQ evaluation identities. The ID score monotonically increases until 10,000 principal components, after which it starts to decrease. We qualitatively visualize this phenomenon in rebuttal Figure 2, where each column represents a fixed seed. 20,000 PCs overfits to the input image and pose, with varying seeds producing only face shots with artifacts which degrade the identity. 1000 PCs underfits the original identity, but shows diversity across different generation seeds. 10,000 PCs fits the identity without overfitting.
>
> **...I am concerned about real-world applications… what are the advantages of w2w compared to existing personalization efforts…”**
>
> We present comparisons against variants of Dreambooth in Table 2 in the main paper, showing how given only a single image, *w2w* can achieve competitive performance with a method which uses multiple images. Following your suggestion, we compare quantitatively against other personalization efforts, Celeb-Basis [1] and IP-Adapter FaceID [2], in rebuttal Table 1. We follow the same evaluation protocol used for Table 2 in the main paper. We further conduct a user study presented in Table 2 of the rebuttal PDF. We show qualitative comparisons against these two methods in rebuttal PDF Figure 6. Details are provided in Global Response 2 and 3.
>
> Overall, across all these metrics, *w2w* performs stronger than Celeb-Basis and IP-Adapter FaceID. Our results indicate that operating in our weight subspace is highly expressive and flexible compared to text embedding space as it is able to faithfully capture nuanced identity without overfitting to the input image. Based on the user study, users found *w2w* generations capture identity better while also generating more diverse images that better align with the prompts. Compared to other learning based approaches like Celeb-Basis, *w2w* is more lightweight as training takes less than a minute with 8GB on a single NVIDIA A100 GPU, while Celeb-Basis takes 21GB and trains a costly MLP in three minutes on a single A100. We present details on the efficiency of *w2w* in Appendix C.
>
> We would like to note that previous personalization methods such as IP-adapter [2] or Celeb-Basis [1] only demonstrate identity inversion. As we mentioned earlier, we create a unified space for identity-encoding weights with applications of inversion, editing, and sampling identity-encoding models.
>
> **“I’m curious if style information…” and “How does w2w perform on multi-concept generation?”**
>
> Thank you for suggesting these creative applications! Since the models used to create *w2w* space are fine-tuned on realistic identities, style information, such as cartoon or painting, would not be encoded in any principal components. However, models from *w2w* space can be merged with other personalized models for multi-concept generation. In Figure 7 of the rebuttal PDF, we train a LoRA with Dreambooth for Pixar style, and merge it with a *w2w* model to showcase multi-concept generation ability.

---

> ### Comment · Reviewer_aBHC · 2024-08-13
> **post rebuttal**
>
> Thanks to the authors for their efforts during the rebuttal. The author's response solved part of my problem. But I still have a lot of concerns that haven't been clarified.
>
> 1. The biggest concern is that collecting a large number of weights to build the w2wk space is far more complex than collecting a large number of images. In addition, in terms of the number of parameters, Celeb-Basis only has 1024 learning parameters.
>
> 2. And for question 1, what guarantees that information like cartoon drawing is not encoded in the principal components? For example, the weight information represents a Buzz Lightyear. Wouldn't Pixar style and Buzz Lightyear images appear at the same time? Then, why is the new image created based on the weight information not affected by the Pixar style?
>
> 3. For question 2, perhaps my question is ambiguous and I am sorry for this, I am actually concerned with the generation of multi-subject, whether the identity information of each person can be maintained when generating multiple characters at the same time (I only saw two case shows in Figure 6 and 7 in the paper).

---

> > ### Author Response · Authors · 2024-08-13
> >
> > We appreciate your response and are happy to clarify further.
> >
> > **“Biggest concern is the collecting a large number of weights…”**
> >
> > We agree that collecting a large number of model weights is more complex than a large number of images. However, the bigger picture of this work is the “idea of using weights as the dataset” (215A) in order to demonstrate "the existence of a LoRA model manifold” (rzPz). As mentioned in the global response, we introduce this idea of a **meta**-latent space **creating new models**, with “broad applications in generative modeling” (wMSF), which we believe is beyond just the personalization application.  We would like to note that although creating this space requires a notable amount of compute, using the space for the variety of applications is lightweight. We discuss these practical details below.
> >
> > **“...in terms of the number of parameters, Celeb-Basis only has 1024 learning parameters.”**
> >
> > While Celeb-Basis uses 1024 coefficients to define an identity in token space, it requires optimizing an MLP at test-time with 525,312 parameters and backpropagation through the text-encoder, leading to \~3 minutes and \~21GB VRAM on a single NVIDIA A100. In contrast, we directly optimize 10,000 coefficients (\~2% of Celeb-Basis's parameters) without text-encoder backpropagation, reducing optimization time to \~1 minute and VRAM usage to \~8GB. Furthermore, our studies (Table 1 and 2 in the rebuttal PDF) show *w2w* offers better identity preservation without overfitting.
> >
> > We would also like to note that Celeb-Basis works in token embedding space, which is already fairly low dimensional. In their case, they use 1024 dimensions to span the original 1536 dimensional space (768 dimensions for first name + 768 for last name), resulting in ~66% of the original dimensionality. In contrast, we compress a much higher dimensional LoRA space (100,000 dimensions) to 10,000 dimensions (10% original dimensionality). The fact that even LoRAs, which are by design low rank, have significantly lower rank semantic subspaces is quite intriguing, potentially inspiring further exploration of interpretable and controllable subspaces in model weights.
> >
> >
> > **“...what guarantees that information like cartoon drawing is not encoded in the principal components?”**
> >
> > When you say, “the weight information represents a Buzz Lightyear,” the weights of the base Stable Diffusion model encodes prior concepts such as “Buzz Lightyear,” “Pixar,” etc, but not the LoRA weights added on top. During Dreambooth fine-tuning, we train LoRAs on realistic human identities, so the only new inserted concept is the identity. We apply PCA to the LoRAs, not the base model, and since there’s no style variation among the encoded identities, no principal components encode style. However, if “style-editing” means generating images of the identity in different styles with prompts, that’s possible because the base model retains prior concepts. For example, in Fig. 7 and Fig. 8, we generate a person as a statue or painting using prompts. Furthermore, in rebuttal PDF Fig. 8, we show that a separately trained style LoRA can be merged with an identity model from *w2w* space.
> >
> > **“... whether the identity information of each person can be maintained when generating multiple characters at the same time…”**
> >
> > Thank you for the clarification. Identities from *w2w* space can be generated alongside existing characters or concepts in the base diffusion model (e.g., the celebrity examples in Figures 6 and 7 of the paper). However, two *w2w* models can't be naively merged by adding weights, as they share the same subspace and will interfere, leading to identity interpolation. Separate LoRAs can be trained to encode identities and then be merged with *w2w*, but they cannot live in the same subspace. This can be enforced using something like [1]. On the other hand, a LoRA encoding another concept such as style can be seamlessly merged as seen in rebuttal PDF Fig. 8. We will discuss this as a limitation in the paper.
> >
> > **Summary**
> >
> > We hope we addressed your main concerns. Overall, we have shown from an efficiency and performance standpoint, that *w2w* performs stronger than Celeb-Basis as well as other baselines. More importantly, as you mention in your original review, our work provides “new ideas and insights into personalized generation models.” We believe that the merits of our work should not be subsumed by Celeb-Basis just because of the personalization application. If that were the case, then works like Dreambooth can be thought of as just Textual Inversion but optimizing model weights instead of token embeddings. But works like Dreambooth provide further insights beyond the application.
> >
> > [1] “Orthogonal Adaptation for Modular Customization of Diffusion Models.” Po et al. 2024

---

> ### Comment · Reviewer_aBHC · 2024-08-14
>
> Thank you for clarifying the doubts. Most of my concerns have been addressed. I will update my rating. However, I still think the use of weights does introduce a large amount of difficulty in preprocessing. Also how to merge multiple LoRa is a big challenge now, and the token-based approach can easily generate multiple roles.

---

### Author Rebuttal · Authors · 2024-08-06

# Global Response:

We sincerely thank the reviewers for their feedback. We are glad that the reviewers found our creation and analysis of a large dataset of model weights “interesting” (abHC, 215A), and the concept of modeling the manifold of diffusion model weights “novel” with “broad applications in generative modeling” (wMSF). Furthermore, we are happy that reviewer wMSF found that our “extensive experiments...demonstrate the effectiveness of the proposed w2w space in generating, editing, and inverting identities”, which further expands the “scope of manifold applications” according to reviewer rzPz. We have uploaded a PDF of visual results to supplement our response to individual reviewers. Below, we address points that may be of interest to all reviewers.

## 1) Summary of Contributions
As pointed out by reviewers 215A and abHC, previous works have explored inversion, editing, or sampling in the context of the **input space** of a model (e.g., latents, text conditioning, noise, images, etc.).  However, *weights2weights* space enables these three applications on the **network weights themselves, producing new models** as illustrated in Figure 2 of the main paper. As such, *weights2weights* space can be thought of as a **meta**-latent space, with “applications analogous to those of a generative latent space – inversion, editing, and sampling – but producing model weights rather than images” (L316-318 in the paper).  We demonstrate how new models can be created simply via linear interpolation within a linear subspace defined by PCA. In our submitted paper, these different models are each encoding a different instance of a visual identity (i.e., a person). We further train linear classifiers using model weights as data to find separating hyperplanes in weight space to edit semantic identity attributes encoded in the model weights. Each edit results in a new model with its original identity changed for some attribute (e.g., adding a beard). In Figure 3 of the rebuttal PDF, we demonstrate that the *weights2weights* idea is not limited to just faces and identities. We hope our work inspires further efforts to discover interpretable and controllable subspaces of model weights.

## 2) Comparison to More Baselines in Identity Inversion
Following the suggestions of the reviewers, we have added quantitative results of *w2w* inversion against more baselines. Our current results in Table 2 of the paper contain comparisons against variants of Dreambooth [4], and we now add a comparison in the rebuttal PDF Table 1 to single-shot personalization methods: Celeb-Basis [1] and IP-Adapter FaceID [2]. We follow the same evaluation protocol as Section 4.4 of the paper, using the same base model. Our results indicate that inversion into *w2w* space achieves better identity preservation. We plan to update Table 2 of the paper with these additional results.

## 3) Identity Inversion User Study
In rebuttal PDF Table 2, we present a two-alternative forced choice (2AFC) user study on the overall quality of generated identities from *w2w* inversion. Twenty users were given ten sets of images. Each set contained a randomly sampled original image of an identity, and then three random images generated using Celeb-Basis [1], IP-Adapter FaceID [2], and *w2w* with the same random prompt. Users were then asked to choose between alternate pairs based on three criteria: identity preservation, prompt alignment, and diversity of generated images. Our results show that users have a strong preference toward *w2w* identities. In Figure 6 of the rebuttal PDF, we present a qualitative comparison of the three methods. We plan to include more such qualitative comparisons in an updated version of the manuscript. We will also include this user study in this update.

## 4) Identity Editing User Study
In rebuttal PDF Table 3, we present a two-alternative forced choice (2AFC) user study to evaluate the quality of identity edits. Twenty-five users were given ten sets of images. Each set contained a randomly sampled original image of an identity, and then an image of that identity edited for an attribute using Concept Sliders [3], *w2w*, and text prompting. Users were then asked to choose between alternate pairs based on three criteria: identity preservation, alignment with the desired edit, and disentanglement. These three criteria are similar to what we measure quantitatively in Table 1 in the paper. Our results show that users have a strong preference toward *w2w* edits. We plan to add this user study in an updated version of the manuscript.



### References:

[1] ”Inserting Anybody in Diffusion Models via Celeb Basis.” Yuan et al. 2023.

[2] “IP-Adapter: Text Compatible Image Prompt Adapter for Text-to-Image Diffusion Models.” Ye et al. 2023

[3] “Concept Sliders: LoRA Adaptors for Precise Control in Diffusion Models.” Gandikota et al. 2023

[4] “DreamBooth: Fine Tuning Text-to-Image Diffusion Models for Subject-Driven Generation.” Ruiz et al. 2023.

[5] “Multi-Concept Customization of Text-to-Image Diffusion.” Kumari et al. 2023.

---

### Author Response · Authors · 2024-08-10

We thank all reviewers for their time and valuable comments on our work. We are grateful that this paper has overall been well-received by the reviewers. As the end of the discussion period is approaching, we kindly ask all reviewers to acknowledge if our rebuttal has addressed their concerns and give us an opportunity to address any further follow-up. Thank you again for your participation.

---

### Decision · Program_Chairs · 2024-09-25

**Decision:**

Accept (poster)

**Comment:**

This paper has received consensus from all reviewers tending to accept the paper with the views ranging from Borderline Accept to Accept. All the reviewers have agreed on the contributions made by the paper and the clarifications by the authors has helped in addressing most of the concerns of the work. In view of the consensus achieved, AC agrees with an acceptance recommendation.